# Argininosuccinic aciduria fosters neuronal nitrosative stress reversed by *Asl* gene transfer

Julien Baruteau[1,2,3], Dany P. Perocheau[1], Joanna Hanley[3,4], Maëlle Lorvellec[3,4], Eridan Rocha-Ferreira[5], Rajvinder Karda [1], Joanne Ng[1,6], Natalie Suff[1], Juan Antinao Diaz [1], Ahad A. Rahim[7], Michael P. Hughes[7], Blerida Banushi[4], Helen Prunty [8], Mariya Hristova[5], Deborah A. Ridout[9], Alex Virasami [10], Simon Heales[3,8], Stewen J. Howe[1], Suzanne M.K. Buckley[1], Philippa B. Mills[3], Paul Gissen [2,3,4] & Simon N. Waddington [1,11]

Argininosuccinate lyase (ASL) belongs to the hepatic urea cycle detoxifying ammonia, and the citrulline-nitric oxide (NO) cycle producing NO. ASL-deficient patients present argininosuccinic aciduria characterised by hyperammonaemia, multiorgan disease and neurocognitive impairment despite treatment aiming to normalise ammonaemia without considering NO imbalance. Here we show that cerebral disease in argininosuccinic aciduria involves neuronal oxidative/nitrosative stress independent of hyperammonaemia. Intravenous injection of AAV8 vector into adult or neonatal ASL-deficient mice demonstrates long-term correction of the hepatic urea cycle and the cerebral citrulline-NO cycle, respectively. Cerebral disease persists if ammonaemia only is normalised but is dramatically reduced after correction of both ammonaemia and neuronal ASL activity. This correlates with behavioural improvement and reduced cortical cell death. Thus, neuronal oxidative/nitrosative stress is a distinct pathophysiological mechanism from hyperammonaemia. Disease amelioration by simultaneous brain and liver gene transfer with one vector, to treat both metabolic pathways, provides new hope for hepatocerebral metabolic diseases.

[1] Gene Transfer Technology Group, Institute for Women's Health, University College London, 86-96 Chenies Mews, London WC1E 6HX, UK. [2] Metabolic Medicine Department, Great Ormond Street Hospital for Children NHS Foundation Trust, London WC1N 3JH, UK. [3] Genetics and Genomic Medicine Programme, Great Ormond Street Institute of Child Health, University College London, 30 Guilford Street, London WC1N 1EH, UK. [4] MRC Laboratory for Molecular Cell Biology, University College London, Gower Street, London WC1E 6BT, UK. [5] Perinatal Brain Repair Group, Institute for Women's Health, University College London, 86-96 Chenies Mews, London WC1E 6HX, UK. [6] Neurology Department, Great Ormond Street Hospital for Children NHS Foundation Trust, London WC1N 3JH, UK. [7] Department of Pharmacology, School of Pharmacy, University College London, 29—39 Brunswick Square, London WC1N 1AX, UK. [8] Department of Paediatric Laboratory Medicine, Great Ormond Street Hospital for Children NHS Foundation Trust, London WC1N 3JH, UK. [9] Population, Policy and Practice Programme, Great Ormond Street Institute of Child Health, University College London, 30 Guilford Street, London WC1N 1E, UK. [10] Histopathology Department, Great Ormond Street Hospital for Children NHS Foundation Trust, London WC1N 3JH, UK. [11] Wits/SAMRC Antiviral Gene Therapy Research Unit, Faculty of Health Sciences, University of the Witswatersrand, Johannesburg, South Africa. Correspondence and requests for materials should be addressed to S.N.W. (email: s.waddington@ucl.ac.uk)

Adeno-associated virus (AAV) vector-mediated gene therapy has achieved promising results in recent clinical trials in liver[1] and neurodegenerative[2] inherited diseases, and has led to the market approval of the gene therapy product to treat RPE65-mediated inherited retinal dystrophy[3]. This success underpins the current interest in this technology, as illustrated by the rapidly expanding number of gene therapy-based clinical trials[4]. Among various AAV capsid variants, AAV serotype 8 (AAV8) has demonstrated its efficacy in liver transduction in preclinical[5] and clinical studies[6]. This serotype also efficiently transduces other tissues including the central nervous system after systemic injection in neonatal mice[7].

As with many liver inherited metabolic diseases, urea cycle defects exhibit a high rate of mortality and neurological morbidity in infancy despite conventional treatment[8]. Successful correction of the urea cycle via AAV-mediated gene therapy has been reported in mouse models of ornithine transcarbamylase deficiency[9], argininosuccinate synthetase deficiency[10], and arginase deficiency[11]. Argininosuccinic aciduria (ASA; OMIM 207900) is the second most common urea cycle defect with a prevalence of 1/218,000 live births[12]. In addition, ASA is an inherited condition proven to cause systemic nitric oxide (NO) deficiency[13] as the disease is caused by mutations in argininosuccinate lyase (ASL), an enzyme involved in two metabolic pathways: (i) the liver-based urea cycle that detoxifies ammonia, a highly neurotoxic compound generated by protein catabolism and (ii) the citrulline-NO cycle, present in most organs, producing NO from L-arginine via nitric oxide synthase (NOS) (Supplementary Fig. 1)[14]. Patients may exhibit an early-onset phenotype with hyperammonaemic coma in the first 28 days of life, or a late-onset phenotype with either acute hyperammonaemia or a chronic phenotype with neurocognitive impairment and progressive liver disease[15]. Compared to other urea cycle defects, ASA patients present with an unusual systemic phenotype, which involves various organs such as brain, liver, kidney, gut and peripheral arteries[15]. The neurological phenotype with a high rate of neurocognitive impairment, epilepsy, ataxia, remains unexplained and contrasts with a lower rate of hyperammonaemic episodes in ASA compared to other urea cycle defects. Various pathophysiological mechanisms have been hypothesised to account for this paradox, including impaired NO metabolism[16]. A hypomorphic $Asl^{Neo/Neo}$ mouse model shows impairment of both urea and citrulline-NO cycles and reproduces the clinical phenotype with impaired growth, multiorgan disease, hyperammonaemia and early death[13]. Common biomarkers of ASA include increased ammonaemia, citrullinaemia, plasma argininosuccinic acid, orotic aciduria and reduced argininaemia[16].

In this study, we characterise the neuropathophysiology of the disease studying the brain of the hyperammonaemic $Asl^{Neo/Neo}$ mouse. We identify features of a cerebral hyperammonaemic disease and a distinct neuronal disease mediated by oxidative/nitrosative stress but not associated with hyperammonaemia. We use a systemic AAV-mediated gene therapy approach as a proof-of-concept study to rescue survival and protect the ASL-deficient brain from both hyperammonaemia and cerebral impaired NO metabolism. To achieve this, we designed a single-stranded AAV8 vector carrying the murine $Asl$ ($mAsl$) gene under transcriptional control of an ubiquitous promoter, the short version of the elongation factor 1 α (EFS) promoter. The vector is administered systemically to adult and neonatal $Asl^{Neo/Neo}$ mouse cohorts.

## Results

**Pathophysiology of the brain disease in ASA**. ASL deficiency causes a systemic NO deficiency due to the loss of a protein complex that facilitates channelling of exogenous L-arginine to NOS[13]. To explore the effect on cerebral NO metabolism, various surrogate biomarkers were investigated. NO concentrations from wild-type (WT) and $Asl^{Neo/Neo}$ mice were evaluated by measurement of nitrite ($NO_2^-$) and nitrate ($NO_3^-$) ions, downstream metabolites of NO, and were found to be significantly increased in $Asl^{Neo/Neo}$ mice in brain homogenates (Fig. 1a), especially in the cerebrum (Supplementary Fig. 2a) and in the diencephalon (i.e. thalamus and hypothalamus) and midbrain (Supplementary Fig. 2b) but not in the hindbrain (i.e. cerebellum, pons, medulla oblongata) (Supplementary Fig. 2c). Similarly, cyclic guanosine monophosphate (cGMP), a signalling pathway physiologically upregulated by NO generated by coupled NOS[17], when measured in brain homogenates, was also found to be increased in $Asl^{Neo/Neo}$ mice (Fig. 1b). Low tissue L-arginine is a consequence of ASL deficiency downstream the metabolic block and can cause NOS uncoupling[18], which leads to the production of reactive oxygen species including superoxide ion ($O_2^-$) or peroxynitrite ($ONOO^-$) with the latter nitrating specific tyrosine residues and generating nitrotyrosine, a marker of oxidative/nitrosative stress[19]. This process can modify the protein structure and function, altering enzymatic activity or triggering an immune response[19]. The detoxification of peroxynitrite by reduced glutathione (GSH) can generate nitrite via the reaction $ONOO^- + 2GSH \rightarrow NO_2^- + GSSG + H_2O$ [20]. Contrasting with increased nitrite/nitrate levels, glutathione concentrations in brain homogenates of $Asl^{Neo/Neo}$ mice were not decreased compared to WT (Supplementary Fig. 2d) although retrospective power calculation showed an under-powered experiment (power of 0.371 with an α type 1 error of 0.05). A sample size calculation for a power of 0.9 necessitated 34 WT and 31 $Asl^{Neo/Neo}$ mice; groups of this size were impossible due to the lack of animals available. In the cortex of WT and $Asl^{Neo/Neo}$ mice, immunostaining against nitrotyrosine was significantly increased in $Asl^{Neo/Neo}$ mice (Fig. 1c) in cells identified as neurons (Fig. 1d). This nitrotyrosine staining was present in most areas of the brain, but highly predominant in the cortex and minimal in the cerebellum (Fig. 1e). Nitrosothiol levels and western blotting against nitrotyrosine and in brain homogenates did not show any difference between WT and $Asl^{Neo/Neo}$ mice (Supplementary Fig. 2e and 2f, respectively). Immunostaining of glial fibrillary acidic protein (GFAP) and CD68, markers of astrocytic and microglial activation, respectively, did not show any difference (Supplementary Fig. 2g). Immunohistochemistry against NOS isoforms showed, in $Asl^{Neo/Neo}$ mice, an increased staining of neuronal NOS (nNOS or NOS1) (Supplementary Fig. 3a) in neurons (Supplementary Fig. 3b), inducible NOS (iNOS or NOS2) (Supplementary Fig. 3c), in neurons (Supplementary Fig. 3d), and endothelial NOS (eNOS or NOS3) (Supplementary Fig. 3e) in endothelial cells (Supplementary Fig. 3f). The brain morphology did not differ between WT and $Asl^{Neo/Neo}$ mice (Supplementary Fig. 4). As measured by TUNEL staining, an increased rate of cell death was observed in the cortex of $Asl^{Neo/Neo}$ mice (Fig. 1f). Collectively these data suggest that a neuronal oxidative/nitrosative stress plays a role in the neuropathology of ASA. However, hyperammonaemia per se can cause brain toxicity through oxidative stress[21]. To investigate whether neuronal oxidative/nitrosative stress is a primary mechanism involved in the phenotype of patients with ASA or is secondary to hyperammonaemia, we designed a gene therapy approach to normalise ammonaemia and target neuronal ASL activity.

**AAV8.EFS.GFP vector targets liver and cerebral neurons**. In order to extend survival and ameliorate the brain phenotype, we designed a vector that was not only able to transduce the liver to correct the defective urea cycle but also the brain, especially neurons. Neonatal CD-1 mice received an intravenous injection

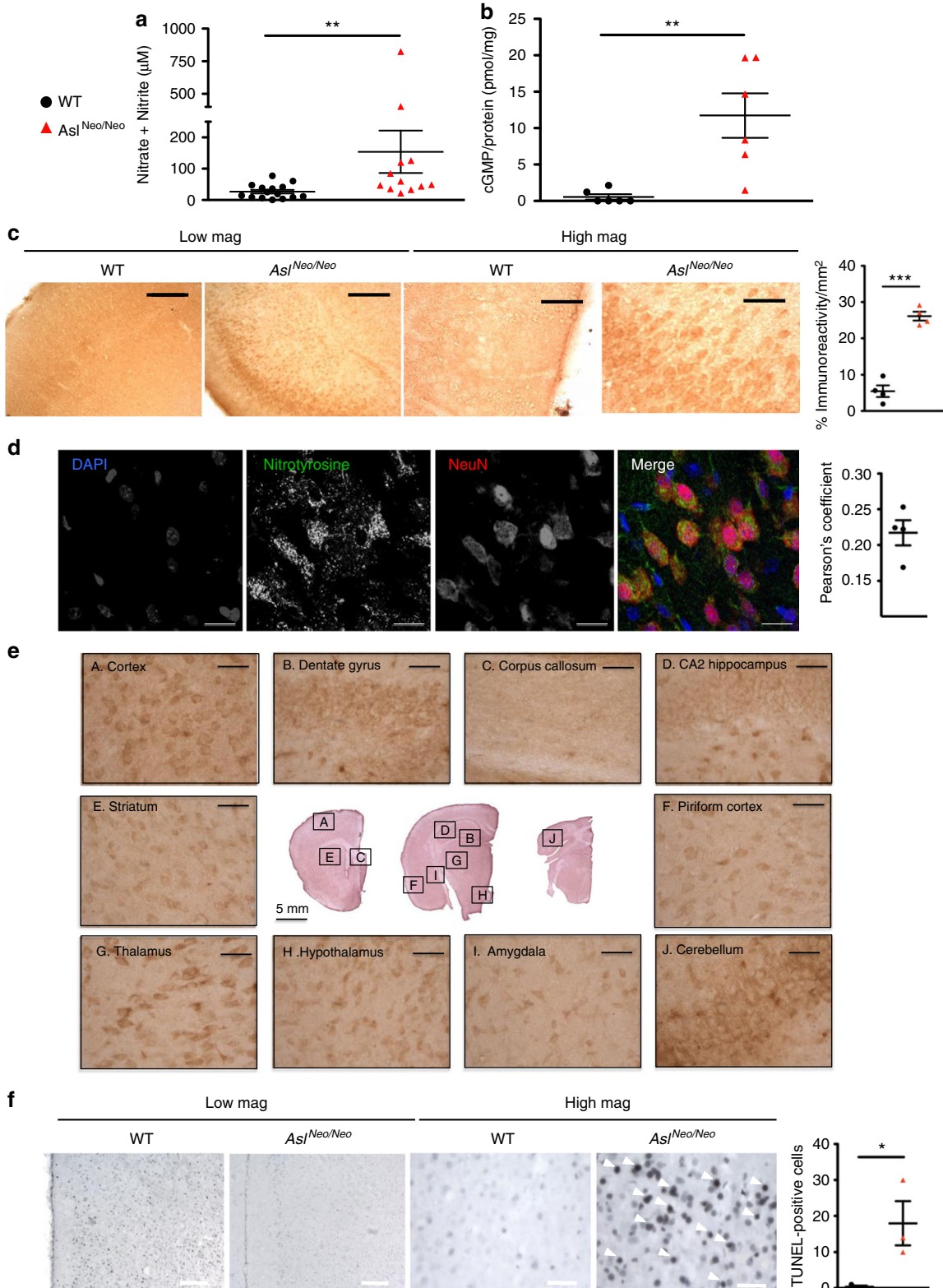

**Fig. 1** Neuronal oxidative/nitrosative stress is a component of the neurological disease in $Asl^{Neo/Neo}$ mice. **a** Nitrite/nitrate levels ($n = 12-15$) and **b** cyclic GMP ($n = 6$) in brain homogenates of 2–4-month-old mice. **c** Nitrotyrosine immunostaining was increased in cortical sections in cells morphologically suggestive as neurons. **d** Neuronal nitrotyrosine staining was confirmed by immunofluorescence ($n = 3$). Colocalisation between nitrotyrosine and NeuN was measured by Pearson's coefficient. **e** Localisation of nitrotyrosine was diffuse, predominant in cortical and subcortical areas and was minimal in the cerebellum. **f** Increased cell death rate in cortex was observed in $Asl^{Neo/Neo}$ mice compared to WT ($n = 4$). Horizontal lines display the mean ± standard error of the mean (SEM). ns = not significant. Unpaired two-tailed Student's $t$ test $*p < 0.05$, $**p < 0.01$, $***p < 0.001$. **a** Graph displays not transformed data. Log-transformed data were used for statistical analysis. Scale bars: **c** low and high magnification: 500 and 125 µm, respectively; **d** 25 µm; **e** 125 µm; **f** low and high magnification: 500 and 125 µm, respectively. Figures show representative images and **d** representative z-projection

of a single-stranded AAV8.EFS.GFP vector ($3.4 \times 10^{11}$ vector genomes/pup) and were culled at 5 weeks of life alongside uninjected control littermates. Fluorescence microscopy revealed green fluorescent protein (GFP) expression in the brain (Fig. 2a) and the liver (Supplementary Fig. 5a). Anti-GFP brain immunostaining showed a transduction prominent in the cortex and decreasing rostro-caudally (Fig. 2b). A pattern of neuronal transduction was suggested by GFP immunostaining (Fig. 2c, d) and confirmed by immunofluorescence (Fig. 2e, f). Anti-GFP immunostaining confirmed a high rate of hepatocyte transduction across the hepatic lobule and other peripheral organs (Supplementary Fig. 5b). Anti-GFP ELISA showed the liver as the main peripheral organ transduced (Supplementary Fig. 5c) with vector genomes detectable 5 weeks after systemic neonatal injection (Supplementary Fig. 5d).

**Impact of gene therapy on the macroscopic phenotype**. A supportive treatment based on a protein-restricted diet and daily intraperitoneal injections of arginine and sodium benzoate was performed in $Asl^{Neo/Neo}$ mice as described in Methods and Supplementary Fig. 6. This improved the survival of untreated $Asl^{Neo/Neo}$ mice (Supplementary Fig. 7a) permitting injection of 30-day-old mice with AAV8 gene therapy (i.e. adult-injected group). A second group of neonatally injected $Asl^{Neo/Neo}$ mice was studied.

Survival was improved significantly in both adult- and neonatally injected groups (Fig. 3a, b). Sustained growth improvement was observed in adult-injected mice (Fig. 3c) with a peak of growth velocity in the 2 weeks following the injection of gene therapy (Supplementary Fig. 7b). In neonatally injected mice, a significant improvement of growth was transiently observed until day 30 (Fig. 3d) consistent with a growth speed similar to WT animals until day 15 (Supplementary Fig. 7b). Later in follow-up, no significant difference of weight was observed between the surviving untreated and neonatally treated $Asl^{Neo/Neo}$ mice (Fig. 3e).

A specific fur pattern with sparse, brittle hair called *trichorrhexis nodosa* was observed in untreated $Asl^{Neo/Neo}$ mice, mimicking symptoms observed in ASA patients[22]. In adult-injected mice, growth and fur pattern dramatically and sustainably improved compared to untreated $Asl^{Neo/Neo}$ mice (Fig. 3f–h). The correction of the fur phenotype was observed within 2 weeks of gene therapy (Supplementary Fig. 8a); the hair shaft was straighter, with a more regular shape, a wider medulla and the restoration of the ability to grow and form physiological tips (Supplementary Fig. 8b–d). Fur aspect and growth were improved transiently in neonatally treated $Asl^{Neo/Neo}$ mice in the first month of life (Fig. 3i–l).

**Long-term improvement of the urea cycle after gene therapy**. At 2 months of age, plasma ammonia was similar to that of WT in both adult- and neonatally injected mice (Supplementary Fig. 9a). Normal ammonia values persisted until sacrifice at 12 months and 9 months after injection in adult- and neonatally injected mice, respectively (Fig. 4a). The plasma concentration of argininosuccinic acid was significantly decreased in adult- but not neonatally injected mice at 2 months of age (Supplementary Fig. 9b). These results were sustained until harvest (Fig. 4b). Similarly, citrulline and arginine plasma concentrations were normalised in adult-injected but not neonatally injected mice (Supplementary Fig. 9c, d). Urinary orotic acid levels were increased significantly in $Asl^{Neo/Neo}$ mice at 10 weeks of age compared with WT mice. Orotic acid concentration was normalised in two adult-injected mice at 10 weeks; however, it did not reach statistical significance in the adult- or the neonatally treated groups (Supplementary Fig. 9e). Plasma alanine

aminotransferase levels were normalised in both adult- and neonatally injected mice (Supplementary Fig. 9f). Liver ASL activity in untreated $Asl^{Neo/Neo}$ mice was $14.5 \pm 4\%$ (range $0-10$ nmol ng$^{-1}$ min$^{-1}$) of WT activity (range $48-68$ nmol ng$^{-1}$ min$^{-1}$). This increased significantly to $47 \pm 33.9\%$ (range $6-53$ nmol ng$^{-1}$ min$^{-1}$) and $18.5 \pm 4.5\%$ (range $8-14$ nmol ng$^{-1}$ min$^{-1}$) in adult- and neonatally injected groups, respectively, at the time of harvest (Fig. 4c). Retrospective power calculation of liver ASL activity between untreated and adult-injected $Asl^{Neo/Neo}$ mice provided a power of 1 with an α type 1 error of 0.05. Anti-ASL liver immunohistochemistry showed a diffuse transduction of cells morphologically identified as hepatocytes, prominent following adult injection and scarce after neonatal injection (Fig. 4d). Quantification of anti-ASL immunohistochemistry showed a significant increase in adult-injected mice (Fig. 4d). Quantitative PCR confirmed greater persistence of vector genomes after adult vs. neonatal gene therapy (Supplementary Fig. 9g).

Haematoxylin and eosin (H&E) staining of liver samples showed vacuolated cytoplasm in untreated $Asl^{Neo/Neo}$ mice; cytoplasmic glycogen deposits were identified by periodic acid Schiff (PAS) staining. This feature was markedly improved following adult, but not neonatal injections (Supplementary Fig. 10).

**Long-term improvement of the NO metabolism in the liver**. Liver NO levels, assessed by nitrite/nitrate levels, were reduced in untreated $Asl^{Neo/Neo}$ mice. These improved in adult-injected but not neonatally injected mice (Supplementary Fig. 11a). Liver glutathione levels were decreased in untreated $Asl^{Neo/Neo}$ mice but did not improved significantly in treated mice (Supplementary Fig. 11b).

**Impact of gene therapy on cerebral NO metabolism**. Cortical ASL enzyme activity in untreated $Asl^{Neo/Neo}$ mice was $14.1 \pm 7\%$ (range $6-33$ pmol ng$^{-1}$ h$^{-1}$) of WT activity (range $71-251$ pmol ng$^{-1}$ h$^{-1}$). In mice injected as adults, this activity was unchanged ($16.2 \pm 5.2\%$ of WT activity (range $0-52$ pmol ng$^{-1}$ h$^{-1}$)) but increased dramatically in mice injected neonatally with $64.8 \pm 34.3\%$ of WT activity (range $30-140$ pmol ng$^{-1}$ h$^{-1}$) being evident at time of culling (Fig. 5a).

To assess the effect of the improved ASL activity on the NO metabolism in brains of neonatally treated $Asl^{Neo/Neo}$ mice, we measured nitrite/nitrate levels. Compared to WT brains, nitrite/nitrate levels were increased in untreated $Asl^{Neo/Neo}$ mice and in adult-injected mice by 3.4 and 2.5 times, respectively, whereas in neonatally injected $Asl^{Neo/Neo}$ mice the levels were not significantly different from WT mice (Fig. 5b). To examine if this decrease in nitrite/nitrate levels in neonatally treated mice was correlated with a modification of the oxidative/nitrosative stress, we quantified cortical nitrotyrosine staining. There was no significant difference between neonatally injected mice and WT mice. In contrast, adult-injected mice and untreated $Asl^{Neo/Neo}$ mice showed a significant increase in the percentage of immunoreactivity (Fig. 5c, d). To assess the NO/cGMP pathway, cGMP levels in brain homogenates were measured. Compared to WT brains, cGMP levels in untreated and adult-treated $Asl^{Neo/Neo}$ mice were significantly higher and normalised in two out of three samples of neonatally treated mice although this did not reach significance (Supplementary Fig. 12).

**Effect of gene therapy on behaviour and cerebral cell death**. Behavioural testing was performed to assess open field exploration. At 3 months of age, there was a significant reduction in the walking distance measured in the untreated $Asl^{Neo/Neo}$ mice,

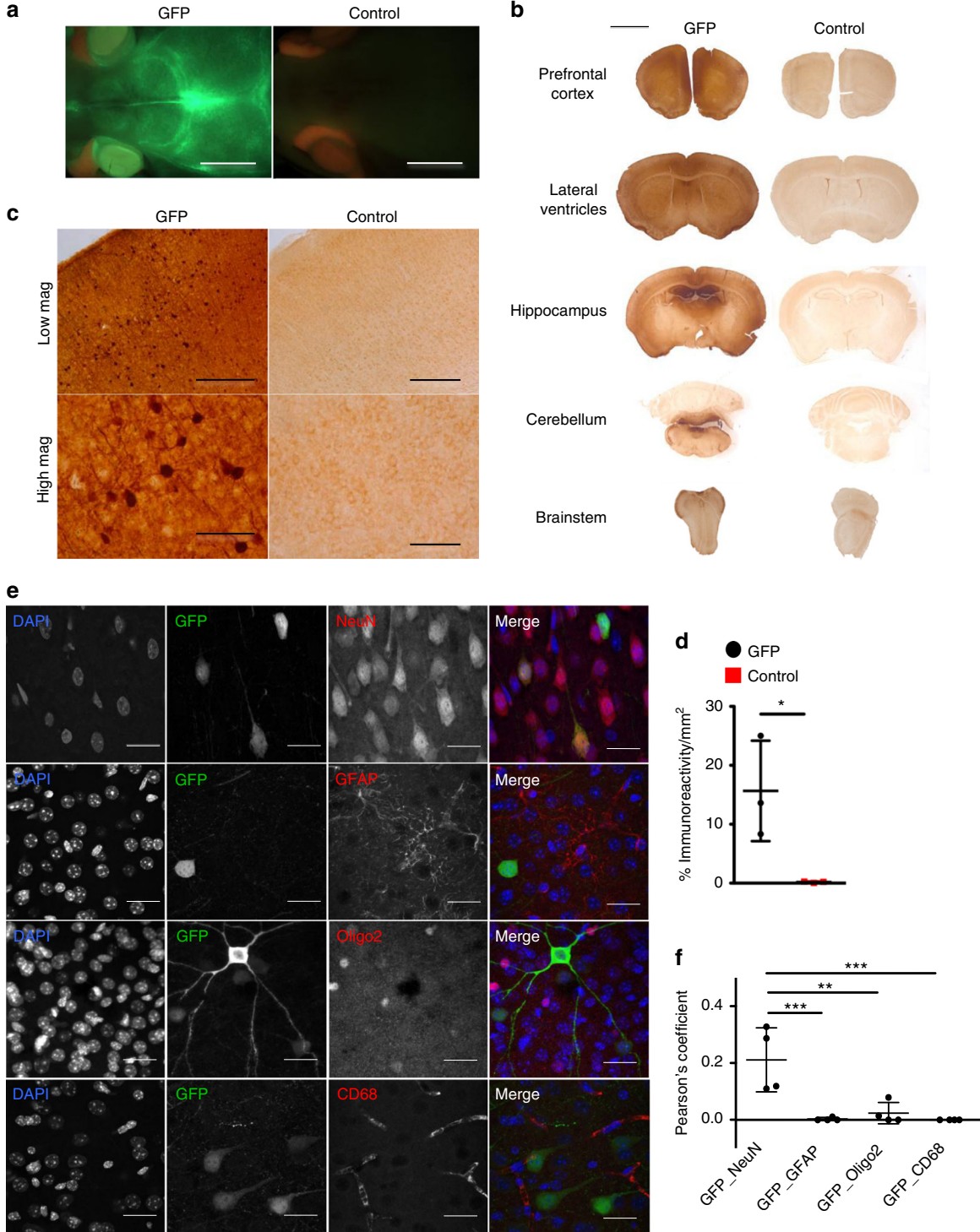

**Fig. 2** Neonatal intravenous injection of AAV8.EFS.GFP enables neuronal transduction. **a** Brain imaged with fluorescence microscope in CD-1 pups injected intravenously with AAV8.EFS.*GFP* (GFP) and uninjected controls. **b**, **c** Representative images of GFP immunostaining in brain at **b** low and **c** at higher magnifications in mice injected with the GFP vector and uninjected controls show a decreasing rostro-caudal gradient with preferential transduction of forebrain and midbrain. **d** Computational quantification of GFP immunostaining showed a significant increase in AAV8.EFS.*GFP*-injected versus uninjected littermates ($n = 4$). **e** Immunofluorescence of cortical staining for DAPI (blue), GFP (green), and NeuN, GFAP, Olig-2, CD68 (red) identifying neurons, astrocytes, oligodendrocytes and microglial cells, respectively. **f** Colocalisation measured by Pearson's coefficient showed a restricted neuronal transduction. Horizontal lines display the mean ± SEM. **d** Unpaired two-tailed Student's *t* test *$p < 0.05$. **f** One-way ANOVA with Dunnett's post-test compared to GFP_NeuN **$p < 0.01$, ***$p < 0.001$. Scale bars: **a**, **b** 5 mm; **c** 500 μm and 125 μm in low and high magnification pictures, respectively; **e** 25 μm. Figures show representative images and **e** representative z-projections for GFP-NeuN and GFP-Olig-2 of four animals. GFP green fluorescent protein

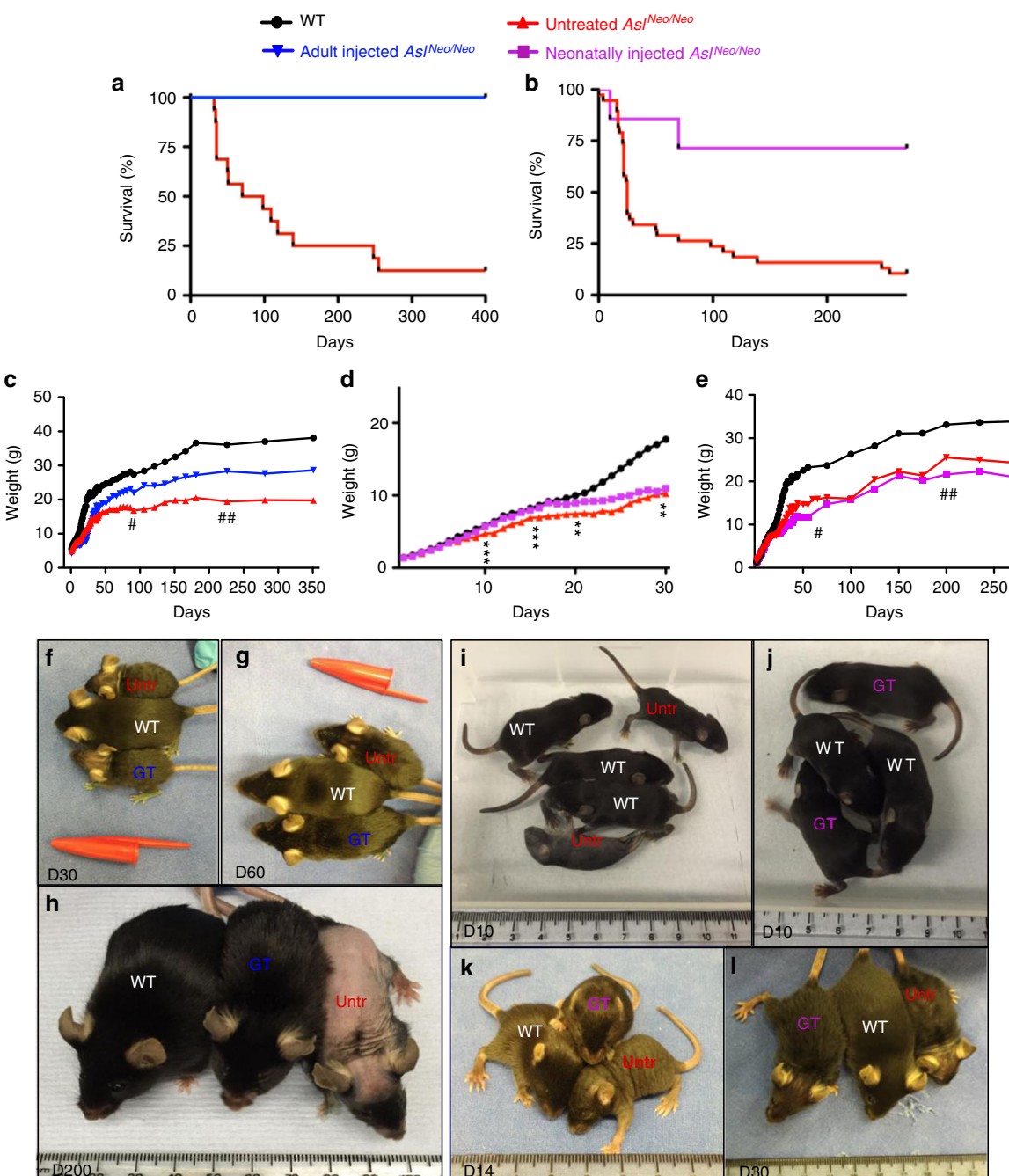

**Fig. 3** Systemic injection of AAV8.EFS.*mAsl* sustainably improves the macroscopic phenotype of *Asl*^Neo/Neo mice injected as adults but only transiently after neonatal injection. **a** Survival curve of adult-injected *Asl*^Neo/Neo mice (*n* = 5/5) compared to untreated *Asl*^Neo/Neo mice (*n* = 2/16); Log rank test *p* = 0.003. **b** Survival curve of neonatally injected *Asl*^Neo/Neo mice (*n* = 5/7) compared to untreated *Asl*^Neo/Neo mice (*n* = 2/21); Log rank test *p* = 0.006. **c** Mean growth of adult-injected *Asl*^Neo/Neo mice (*n* = 5) compared to WT (*n* = 11) and untreated *Asl*^Neo/Neo mice over 12 months (*n* = 19). **d, e** Mean growth of neonatally injected *Asl*^Neo/Neo mice compared to WT (*n* = 31) and untreated *Asl*^Neo/Neo mice (*n* = 41) **d** during the first month (*n* = 13) and **e** over 9 months (*n* = 7). **f**–**h** Images of WT, untreated *Asl*^Neo/Neo (Untr), and adult-injected *Asl*^Neo/Neo mice with gene therapy (GT). **i**–**l** Images of WT, untreated *Asl*^Neo/Neo (Untr) and neonatally injected *Asl*^Neo/Neo mice (GT). Unpaired two-tailed Student's *t* test **\*\****p* < 0.01, \*\*\**p* < 0.001, neonatally injected vs. untreated *Asl*^Neo/Neo mice. # 30% and ##<15% of untreated *Asl*^Neo/Neo mice still alive; scale bar **f**–**l** 1 cm

whereas an improvement was seen in both adult- and neonatally injected groups (Fig. 6a). Performance with an accelerating rotarod at the same age was tested and showed a significant reduction in untreated *Asl*^Neo/Neo mice but not significantly different from WT in both adult- and neonatally injected groups (Fig. 6b). This is even more remarkable, when the fact that heavy mice can perform worse than light ones[23] is taken into consideration.

Cell death was assessed by TUNEL staining and was found to be significantly increased in the cortex of untreated *Asl*^Neo/Neo mice compared to WT. Cell death was reduced in adult-injected compared to untreated *Asl*^Neo/Neo mice. In neonatally injected mice, this parameter was further improved compared to adult-injected mice with no significant difference compared to WT mice (Fig. 6c, d).

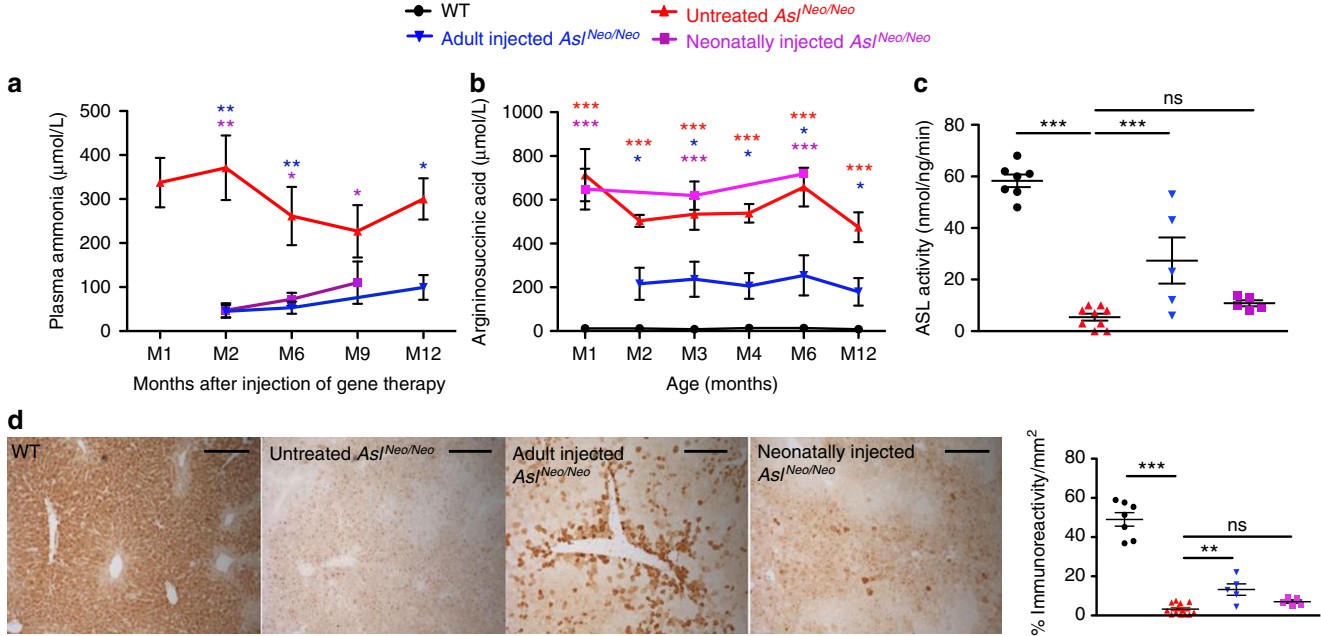

**Fig. 4** AAV8.EFS.*mAsl* controls hyperammonaemia and restores a functional urea cycle. Concentration of (**a**) plasma ammonia and **b** argininosuccinic acid in dried blood spots overtime in WT, untreated, adult-injected and neonatally injected *Asl*$^{Neo/Neo}$ mice. **c** Liver argininosuccinate lyase (ASL) activity at culling. **d** Representative images of ASL-stained sections of liver in WT, untreated, adult-injected and neonatally injected *Asl*$^{Neo/Neo}$ mice (*n* = 5) and computational quantification of ASL immunostaining. Horizontal lines display the mean ± SEM. One-way ANOVA with Dunnett's post-test compared to **a**, **b** WT and **c**, **d** untreated *Asl*$^{Neo/Neo}$. ns—not significant; *$p < 0.05$; **$p < 0.01$; ***$p < 0.001$. **d** Scale bars 500 μm. WT *n* = 13–20 (littermates aged 9 or 13 months); untreated *Asl*$^{Neo/Neo}$ *n* = 10–12 (littermates aged 1–13 months as less than 10% of the animals had survived at the end of the study); adult-injected *Asl*$^{Neo/Neo}$ *n* = 4–5 (aged 13 months); neonatally injected *Asl*$^{Neo/Neo}$ *n* = 3–5 (aged 9 months)

## Discussion

This study provides new insight into the pathophysiology of the brain disease in ASA. This work highlights the role of a neuronal disease not caused by hyperammonaemia, which supports repeated clinical observations that ASA patients present poor neurocognitive performance even without hyperammonaemia. Systemic AAV-mediated gene therapy provides proof-of-concept of hepatocerebral phenotypic correction of the hypomorph *Asl*$^{Neo/Neo}$ mouse model, which open avenues for gene therapy in ASA.

Compared to other urea cycle defects, the neurological disease in ASA is a paradox since low rates of hyperammonaemic decompensation are accompanied by high rates of neurological complications, including neurocognitive delay, abnormal neuroimaging, epilepsy and ataxia[16]. We characterised the previously unreported neuropathology in the *Asl*$^{Neo/Neo}$. NO plays a complex and ambiguous role in the brain, involved in both inflammation-related neurotoxicity and cGMP-mediated neuroprotection[24]. Cerebral NO levels (assessed by nitrite/nitrate levels) were increased in some brain areas (cerebrum, diencephalon, midbrain) but not in the hindbrain. Increased NO and cGMP in the brain of *Asl*$^{Neo/Neo}$ mice suggests a persisting physiological upregulation of the NO/cGMP pathway, which is observed with appropriate coupling of NOS[17]. This has been observed in animal models and patients with hyperammonaemia caused by hepatic encephalopathy with increased cortical guanylate cyclase activity[25]. Different alterations of NO metabolism in specific brain areas might explain why the analysis of whole brain lysates might reveal no differences in this pathway. For instance, Erez et al. found no difference in nitrite and nitrosothiols levels when comparing brain lysates from WT and *Asl*$^{Neo/Neo}$ mice[13]. However, previous studies in *Asl*$^{Neo/Neo}$ mice have demonstrated the uncoupling of NOS likely promoted by low tissue L-arginine content, which is associated with the increase of

systemic biomarkers of oxidative stress[18]. NOS uncoupling causes oxidative/nitrosative stress in vitro with the production of peroxynitrite, which in turn contributes to decrease of antioxidants, inhibition of the mitochondrial respiratory chain, opening of the permeability transition pore and cell death[20]. This is consistent with our observation of increase of both neuronal nitrotyrosine staining and nitrite/nitrate levels. In the brain, hyperammonaemia increases nNOS and iNOS-mediated NO synthesis via an increase in extracellular glutamate, and activation of the glutamate-NO-cGMP pathway via *N*-methyl-D-aspartate receptors[26]. High levels of NO can lead to oxidative stress[21]. In our study, correction of hyperammonaemia alone did not modify the oxidative/nitrosative stress, suggestive of an independent brain-specific causative mechanism. A measurable reduction in cell death in the cortex was observed when neuronal ASL activity is restored. Neurons are more vulnerable to oxidative stress than astrocytes in vitro, as they cannot overexpress γ-glutamyl transpeptidase to replenish their intracellular glutathione content[27]. Therefore, they rely on the paracrine glutathione supply from astrocytes when exposed to reactive nitrogen species and oxidative stress[27]. This might explain the neuronal staining observed for nitrotyrosine and the efficacy of neuronal-targeted gene therapy. This nitrosative stress caused by peroxynitrite, generating nitrotyrosine, has been implicated previously in the pathophysiology of various neurodegenerative diseases: Parkinson's disease[28], Alzheimer's disease[29] and amyotrophic lateral sclerosis[30]. In urea cycle defects, a neuronal disease caused by oxidative stress as a consequence of low tissue arginine has been hypothesised as playing a role in the brain pathophysiology[31]. Although the precise biochemical mechanisms regulating NO metabolism in different cerebral cell types in ASA remain elusive, compelling evidences from this study and the literature[16,22,26,31–33] support the coexistence of NOS coupling and uncoupling in the brain *Asl*$^{Neo/Neo}$ mice accounting for both

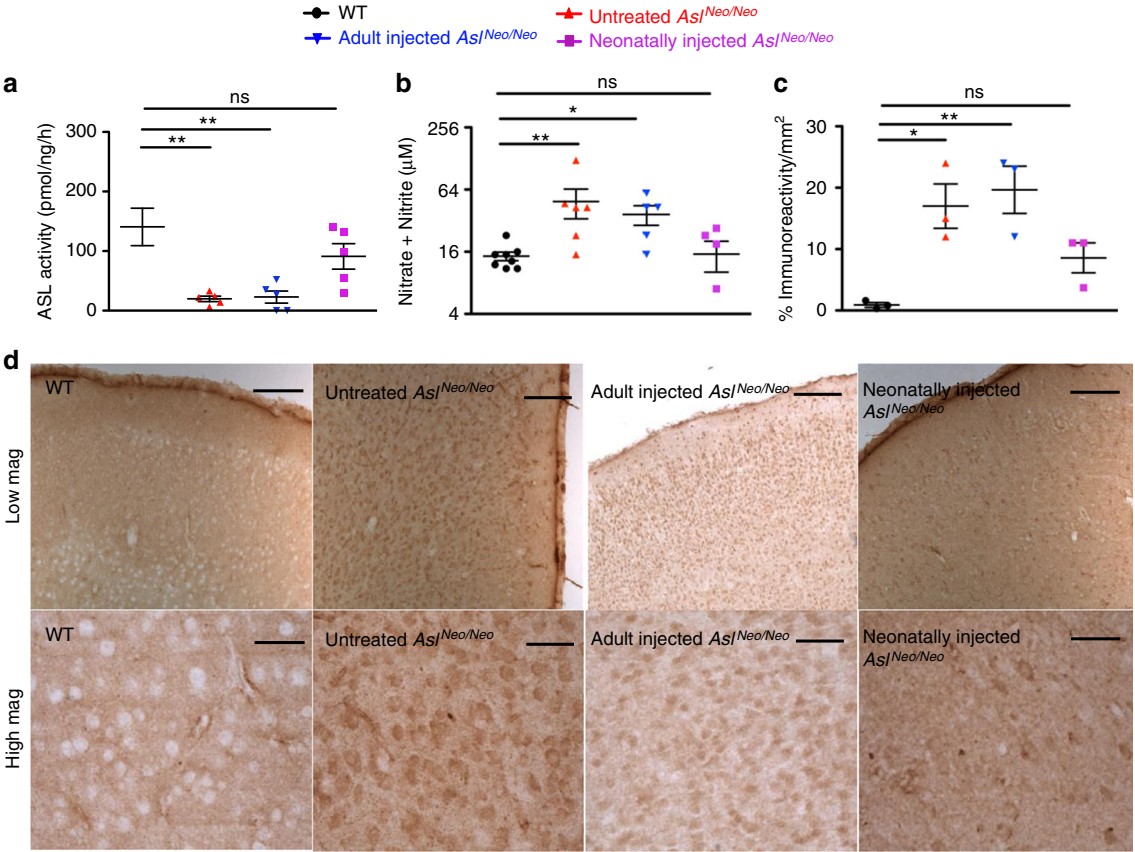

**Fig. 5** Systemic neonatal but not adult injections of AAV8.EFS.*mAsl* exhibited long-term correction of the impaired NO metabolism in the brain. **a** Cortical argininosuccinate lyase (ASL) activity. **b** Levels of nitrite/nitrate levels in brain homogenates. **c** Computational quantification of nitrotyrosine immunostaining. **d** Representative images of nitrotyrosine-stained sections of cortex in WT, untreated, adult-injected and neonatally injected *Asl*[Neo/Neo] mice (n = 3). Horizontal lines display the mean ± SEM. **a−c** One-way ANOVA with Dunnett's post-test compared to WT: ns—not significant; *p < 0.05; **p < 0.01. **b** Graph displays not transformed data. Statistical analysis used log-transformed data. WT n = 6–8 (littermates aged 9 or 13 months); untreated *Asl*[Neo/Neo] n = 5–6 (littermates aged 1–13 months as less than 10% of the animals had survived at the end of the study); adult-injected *Asl*[Neo/Neo] n = 5 (aged 13 months); neonatally injected *Asl*[Neo/Neo]

the physiological glutamate-NO-cGMP pathway and nitrosative/oxidative stress, respectively (Fig. 7) as observed in Alzheimer's disease[34]. No evidence of neuroinflammation was observed in astrocytes and microglial cells, as assessed by GFAP and CD68 immunohistochemistry, suggesting that these cell types are not primarily involved. NO supplementation was reported in one ASA patient[18]: this dramatically normalised refractory arterial hypertension, likely by restoring impaired vasoregulation and correcting peripheral NO deficiency. This was associated with improvement in some neurocognitive tests although the IQ was unchanged. An improved regulation of the cerebral blood flow might explain some of the cognitive amelioration, although the authors did not exclude biased results by practice effect. This does not discount the role of a nitrosative stress-related cerebral disease and an ongoing clinical trial assessing NO supplementation in the neurocognitive function of ASA patients (NCT 03064048) might provide further understanding of the neurological phenotype of ASA. Thus neuronal oxidative/nitrosative stress seems to play a key-role in the ASA brain disease.

In murine models, AAV8 is known for its ability to widely transduce the brain after intracranial administration[32,33]. After systemic injection, most of the organs and especially the liver are successfully targeted[35] but the neurotropism is influenced by the age of infusion and the dose of vector administered. For instance, successful brain transduction with AAV8 and a CMV promoter

after systemic delivery has been previously reported no later than day 14 of life (1.5×10$^{11}$ vg per mouse)[36]. However brain transduction was barely detectable in adult mice after a similar experiment (1×10$^{11}$ vg per mouse)[35]. Increasing the dose of vector improved brain transduction in adult mice. Indeed intravenous injection in adult mice with AAV8 and EF1α promoter showed mild brain cell transduction at 3×10$^{11}$ to 1.8×10$^{12}$ vg per mouse[37], but widespread neuronal and astrocytic transduction at 7.2×10$^{12}$ vg per mouse (approximately 2.9×10$^{14}$ vg kg$^{−1}$)[38]. The transient ability for AAV vectors to cross an immature blood–brain barrier in the neonatal mouse brain is not well understood and could be due to immaturity or receptor-mediated transcytosis[36,39]. The age at injection does not only allow an increased brain transduction but influences the cell types transduced. A predominant neuronal transduction is observed when the vector is administered during the first 48 h of life whereas a preferential astrocytic transduction is noticed from day 3 onwards[33]. These observations were made with an AAV9 serotype, in mice[40,41] and non-human primates[42], with a long-standing transgene expression of up to 18 months in mice after a single systemic neonatal injection[36]. For this study, the choice of EFS promoter was made on the basis of various advantages: relatively ubiquitous expression and strong promoter activity[43], resistance to silencing[44], reduced potential risk of insertional mutagenesis compared to other ubiquitous promoters[45], already used in clinical trials[46]. In the brain, the AAV8.

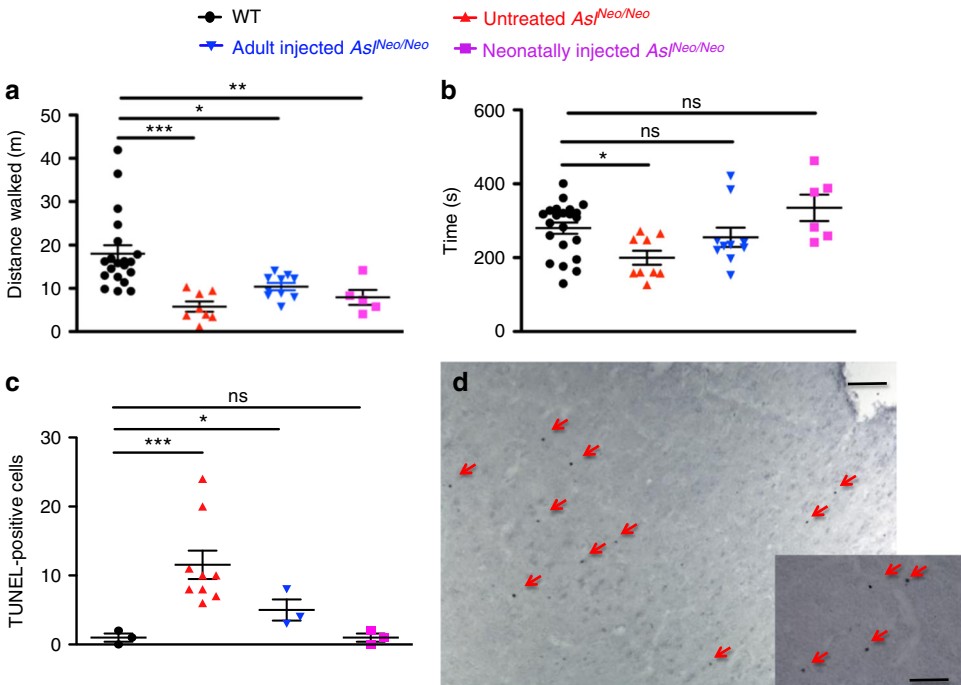

**Fig. 6** AAV-mediated gene therapy improves behavioural testing and is inversely correlated with cortical neuronal loss in treated animals. **a** Open field test and **b** accelerating rotarod performed in 2-month-old mice. **c**, **d** Apoptotic cells at time of culling in WT (aged 9–13 months), untreated (aged 21 days – 13 months), adult-injected (aged 13 months) and neonatally injected $Asl^{Neo/Neo}$ mice (aged 9 months). **c** Cell counting in cortex. **d** Representative cortex image of an untreated 25-day-old $Asl^{Neo/Neo}$ mouse symptomatic of hyperammonaemia and culled for humane endpoint. Horizontal lines display the mean ± SEM. **a**, **b** One-way ANOVA with Dunnett's post-test compared to WT: ns—not significant, $*p < 0.05$, $**p < 0.01$, $***p < 0.001$. **c** Graph displays not transformed data. Statistical analysis used log-transformed data. WT $n = 20$–22; untreated $Asl^{Neo/Neo}$ $n = 8$–9; adult-injected $Asl^{Neo/Neo}$ $n = 5$; neonatally injected $Asl^{Neo/Neo}$ $n = 5$. Scale bars: 500 μm and 125 μm in main and inset pictures, respectively

EFS.GFP vector targeted mainly neurons due to its promoter specificity[47] and the neonatal timing of injection[33].

Long-term correction of ammonaemia levels was observed in mice injected as adults or neonates demonstrating successful restoration of ureagenesis. Adult-injected mice exhibited a more complete correction with prolonged improvement of other typical features of the deficient urea cycle: growth, fur, blood amino acid concentrations and in some, orotic aciduria and liver intracellular glycogen deposits. The correction of the phenotype correlated with the liver ASL activity. AAV vectors deliver nonintegrating transgene copies, persisting as episomes in the transduced cell. The transient nature of the majority of the metabolic effects after neonatal injection is likely caused by a loss of transgene vector genomes in the rapidly growing liver during the first weeks of life and is consistent with previous studies[9,48]. These results are consistent with previous experiments using AAV-derived vectors in murine models of urea cycle defects[9,10]. In this study, an increase from 14.5 to 18% of WT liver ASL activity was observed 9 months after neonatal injection. This provided a persistent correction of ammonaemia but was not sufficient to normalise other biochemical parameters of the disease (plasma amino acids, orotic aciduria). In ASA, the increased urine secretion of the argininosuccinic acid that removes two nitrogen moieties may explain the reduced tendency to develop hyperammonaemic episodes compared to proximal urea cycle defects[14]. AAV-mediated correction of other models of urea cycle disorders has shown that a small improvement (approximately 3%) in liver enzyme levels and ureagenesis can restore survival and improve ammonia levels[49]. Controlling orotic aciduria in the $Spf^{ash}$ mouse model of ornithine transcarbamylase deficiency however required five times more vector compared to that necessary to normalise ammonaemia[50]. In that respect, our study provides a

hierarchization in the significance of biomarkers, according to the ASL residual activity in ASA. Plasma amino acids and urine orotic acid required a liver ASL activity of >18% for normalisation whereas ammonaemia was seen to normalise when ASL activity was only 14.5–18% of WT activity. However these figures might be biased by the persistence of the nonintegrating transgene delivered by AAV vector. As reported previously in shRNA-induced hyperammonaemic $Spf^{ash}$ mice[50], the AAV-encoded enzymatic activity required to normalise ammonaemia might be higher than the endogenous residual activity required in a non-hyperammonaemic subject. Indeed the reduced transgenic expression from nonintegrated episomes compared to endogenous chromosomal alleles has been suspected recently from results of a liver-directed clinical trial[51]. The fur phenotype observed in ASL- and argininosuccinate synthase-deficient mice is likely to be caused by hypoargininaemia as arginine represents up to 10% of hair composition[15]. The long-term phenotypic improvement of the fur in adult-injected mice is consistent with the improved plasma arginine levels. The vector dose administered intravenously ($3 \times 10^{11}$ vg in a 1.5 gram-weighted neonatal mouse) is similar to intravenous doses ($2 \times 10^{14}$ vg kg$^{-1}$) recently published in a successful clinical trial delivering intravenous AAV9 gene therapy for spinal muscular atrophy[52].

A small but significant increase of the nitrite/nitrate levels was observed in the livers of adult-injected mice thereby suggesting that restoration of ASL activity had a positive effect on the function of both urea and citrulline-NO cycles. Reflecting increased cellular levels of oxidative stress, reduced glutathione levels are decreased in untreated $Asl^{Neo/Neo}$ mice but were not significantly improved after gene therapy suggesting that additional factors could play a role such as the rate of hepatocyte transduction. While systemic NO deficiency[13] and increased

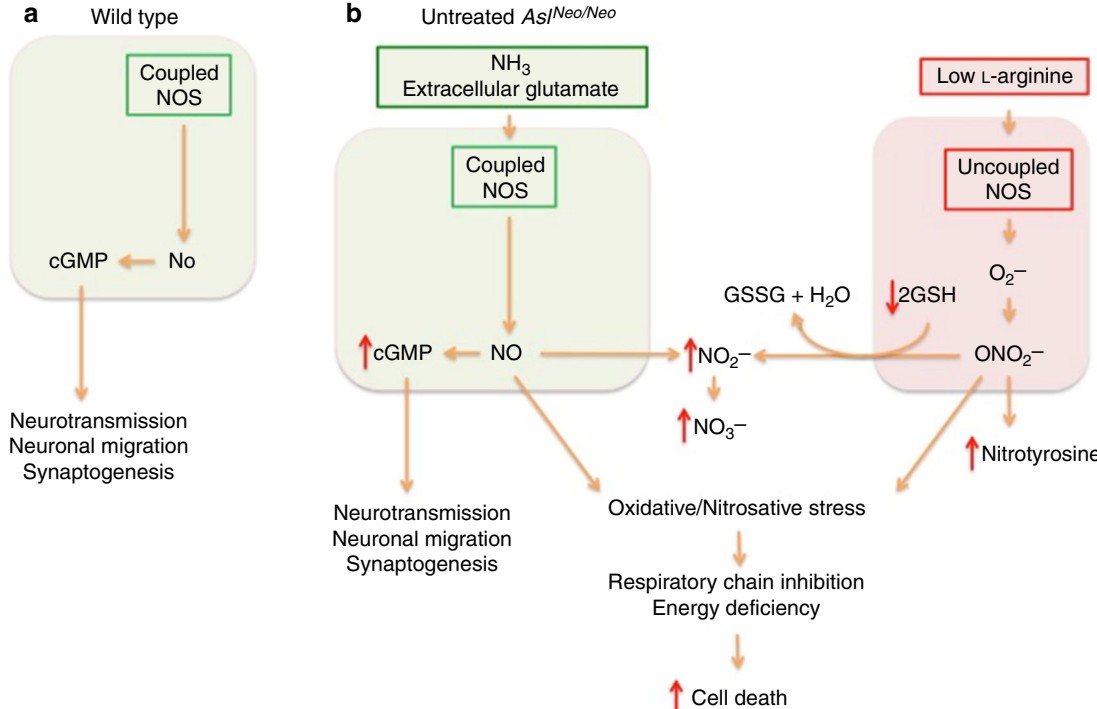

**Fig. 7** Proposed pathophysiological mechanisms of the brain disease in $Asl^{Neo/Neo}$ mice. **a** Residual coupled NOS produces NO with increased cGMP production. **b** Low L-arginine causes NOS uncoupling and produces superoxide ions ($O_2^-$), which will generate the production of peroxynitrite ($ONOO^-$). $ONOO^-$ causes oxidative/nitrosative stress with production of nitrotyrosine and nitrite ($NO_2^-$) after detoxification of $ONOO^-$ by reduced glutathione (GSH). Oxidative/nitrosative stress will impair the respiratory chain, alter the energy production of the cell and lead to cell death. Neurons are particularly vulnerable as they cannot increase their production of glutathione adequately, and rely on an astrocytic supply. **b** Red arrows represent modified measured parameters in $Asl^{Neo/Neo}$ compared to WT mice. This figure relies on data provided in this study and in the literature[13,18,21,25,27] and discussed in the manuscript. cGMP cyclic guanosine monophosphate, GSH reduced glutathione, GSSG oxidised glutathione, NO nitric oxide, $NO_2^-$ superoxide ion, $ONO_2^-$ peroxynitrite, NOS nitric oxide synthase

oxidative stress[18] have been described previously, it was shown that long-term correction of neuronal ASL achieved a marked decrease of the cortical oxidative/nitrosative stress, independent of improvement in ammonia levels.

ASA patients are at high risk of developing neurological complications even if hyperammonaemic episodes do not occur[16]. Our data provide further evidence for a neuronal disease with oxidative/nitrosative stress independent of ammonaemia, and illustrates the pathophysiological importance of disturbed NO metabolism in the ASA brain. Any therapy aiming to preserve the neurological status of ASA patients needs to protect the brain from two potential insults, hyperammonaemia and disturbed cerebral NO metabolism. Current best-accepted therapeutic guidelines aim to normalise ammonia levels and correct hypoargininaemia, but do not target systemic NO imbalance[53]. Similar to liver transplantation[54], any novel therapeutic approach targeting only hepatocytes will cure the urea cycle defect but will not correct the symptoms related to ASL deficiency in extra-hepatic tissues, especially the brain[18], and will be unlikely to improve the long-term neurological outcome of these patients. This study provides proof-of-concept for phenotypic correction of the $Asl^{Neo/Neo}$ mouse model using AAV technology at clinically relevant doses. Acknowledging that clinical translation will require optimisation of the current vector, these promising results raise the possibility of combining two sequential systemic injections: (i) a first early (neonatal) injection of a gene therapy vector that would transiently restore the urea cycle in the liver and will transduce neurons to modify the long-term natural course of the neuronal disease, and (ii) a second injection in infancy or adulthood targeting the liver for long-term correction of the urea

cycle. The potential for humoral immune response generated by the first AAV injection will need to be considered for the second injection[55]. It is possible however, similarly to what is reported for neonatal rodents[56], that the immaturity of the immune system in humans at the time of neonatal injection might prevent the induction or diminish the magnitude of humoral response against the AAV capsid[57]. An alternative AAV capsid, which does not cross-react with neutralising antibodies developed, might be a valid option[58]. Several inherited metabolic diseases with hepato-cerebral phenotype might benefit from a similar dual targeting approach such as mitochondrial diseases caused by nuclear genetic defects (e.g. *POLG1*, *MPV17*, *DGUOK* genes) and some lysosomal storage disorders (e.g. neuronopathic Gaucher disease, mucopolysaccharidosis type I and II). Depending on the pathophysiology of the disease, specific brain cell-types can be selectively targeted in modifying either promoter and/or age at injection[32,33].

## Methods
**Animals.** The $Asl^{Neo/Neo}$ mice (B6.129S7-$Asl^{tm1Brle}$/J) were purchased from Jackson Laboratory (Bar Harbor, ME). For the initial experiments studying the neurological phenotype, WT and $Asl^{Neo/Neo}$ littermates were maintained on standard rodent chow (Harlan 2018, Teklab Diets, Madison, WI; protein content 18%) with free access to water. For gene therapy experiments, all WT and $Asl^{Neo/Neo}$ mice were started on a supportive treatment including a reduced-protein diet (5CR4, Labdiet, St. Louis, MO; protein content 14.1%) from day 15 to day 50. Untreated and adult-injected $Asl^{Neo/Neo}$ mice received daily intraperitoneal injections of sodium benzoate (0.1 g kg$^{-1}$ d$^{-1}$) and L-arginine (1 g kg$^{-1}$ d$^{-1}$) from day 10 to day 30 (Supplementary Fig. 6). CD-1 mice were purchased from Jackson Laboratory (Bar Harbor, ME). For all experiments, WT and $Asl^{Neo/Neo}$ littermates were housed in the same cages. Injected pups cannibalised by the dam were excluded from the analysis. Cross-fostering with CD-1 dams overcame this issue. Mouse experiments

were approved by institutional ethical review and performed under UK Home Office licences 70/6906 and 70/8030.

**Experimental design**. Two groups of $Asl^{Neo/Neo}$ mice were treated with gene therapy at either adult or neonatal age and compared to WT or untreated $Asl^{Neo/Neo}$ littermates. Adult-treated and untreated $Asl^{Neo/Neo}$ littermates received daily supportive treatment (Supplementary Fig. 6) allowing improved survival until day 30 (Supplementary Fig. 7a) but with no improvement on growth. Some $Asl^{Neo/Neo}$ mice received a single intraperitoneal injection of AAV8.EFS.*mAsl* ($2.5 \times 10^{11}$ vg per mouse). Arginine and sodium benzoate injections were withdrawn the following day for all $Asl^{Neo/Neo}$ mice that had received gene therapy (adult-injected $Asl^{Neo/Neo}$ mice) or not (untreated $Asl^{Neo/Neo}$ mice). These mice were monitored until sacrifice at 12 months after injection. Neonatally treated $Asl^{Neo/Neo}$ mice were injected within 24 h of birth with a single intravenous injection of AAV8.EFS.*mAsl* ($3.2 \times 10^{11}$ vg per mouse). The neonatally injected and untreated littermates mice did not receive any supportive treatment and were monitored until sacrifice at 9 months after injection. All untreated $Asl^{Neo/Neo}$ mice were analysed together.

**Behavioural testing**. *Rotarod*: After a period of acclimatisation including 3 trials per day for 3 consecutive days, the test was performed on a Rotarod LE 8200 (Panlab, Harvard apparatus, Cambridge, UK) with 3 attempts per day for 5 consecutive days. The latency to fall from the rod under continuous acceleration from 4 to 40 rpm over 5 min was recorded.

*Open field test*: The animal was placed in the centre of a plastic box ($40 \times 40$ cm floor) and video-recorded for 5 min. Computational analysis of the distance walked was performed automatically using the MouseLabTracker v0.2.9 application on Matlab software (Mathworks, Natick, MA, USA)[59].

**Reagents and antibodies**. All chemicals were from Sigma-Aldrich unless stated otherwise. Antibodies used in this study include the following: rabbit polyclonal anti-GFP (1:10,000 for immunohistochemistry (IHC), 1:4000 for immuno-fluorescence (IF); Ab290, Abcam, Cambridge, UK), chicken polyclonal anti-GFP (1:1000 for IF; Ab13970, Abcam, Cambridge, UK), mouse polyclonal anti-nitrotyrosine (1:800 for IHC; 06-284, Merck Millipore, Temecula, CA, USA), mouse monoclonal anti-nitrotyrosine clone 1A6 (1:100 for IF and western blot (WB); 05-233, Merck Millipore, Temecula, CA, USA), rabbit polyclonal anti-nNOS (1:200 for IHC, 1:100 for IF; bs10197R, Bioss antibodies, Woburn, MA, USA), rabbit polyclonal anti-iNOS (1:500 for IHC, 1:100 for IF; NBP1-50606, Novus Biologicals, Abingdon, UK), mouse purified anti-eNOS (1:300 for IHC, 1:100 for IF; 610296, BD Transduction Lab), mouse monoclonal anti-GFAP (1:500 for IHC, 1:250 for IF; MAB3402, Merck Millipore, Temecula, CA, USA), rat monoclonal anti-GFAP (1:250 for IF, 13-0300, ThermoFisher Scientific, Rockford, IL, USA), rat monoclonal anti-CD68 (1:100 for IHC and IF; MCA1957, Bio-Rad, Oxford, UK), rabbit polyclonal anti-ASL (1:1000 for IHC; Ab97370, Abcam, Cambridge, UK), rabbit polyclonal anti-Olig-2 (1:100 for IF; Ab9610, Abcam, Cambridge, UK), mouse monoclonal anti-NeuN (1:1000 for IF; Millipore, Billerica, MA, USA), rabbit monoclonal anti-NeuN (1:1000 for IF; Ab177487, Abcam, Cambridge, UK), mouse monoclonal anti-CD31 clone 390 (1:20 in IF, ThermoFisher Scientific, Rockford, IL, USA) and goat anti-rabbit secondary antibody (1:1000 for IHC and WB; Vector, Burlingame, CA, USA). All secondary antibodies used for immuno-fluorescence were Alexa Fluor conjugates diluted at 1:500 (Life Technologies). 4′,6-diamidino-2-phenylindole (DAPI) was used for nucleus counterstaining.

**Genotyping**. DNA extraction from tail or ear clips was performed by adding 25 mM NaOH, 0.2 mM EDTA adjusted to pH 12 prior to heating the sample at 95 °C for 10 min. An equal volume of 40 mM Tris (adjusted to pH 5) was then added. DNA was amplified using a Taq DNA Polymerase PCR kit (Peqlab, Germany) according to the manufacturer's instructions using the following primers: 5′-GGTTCTTGGTGCTCATGGAT-3′ (sense), 5′-GCCAGAGGCCACTTGTG TAG-3′ (WT, antisense) and 5′-CATGACAGCTCCCATGAAGA-3′ ($Asl^{Neo/Neo}$ mice, antisense) provided by Jackson Laboratory (Bar Harbor, ME). Amplification conditions were 95 °C for 10 min then 40 cycles of 94 °C for 30 s, 63 °C for 30 s, 72 °C for 1 min.

**Cell culture**. Human embryonic kidney (HEK) 293 cells were maintained in Dulbecco's modified Eagle medium (Gibco, Invitrogen, Grand Island, NY) supplemented with 10% (vol/vol) fetal bovine serum (JRH, Biosciences, Lenexa, KS) and maintained at 37 °C in a humidified 5% $CO_2$-air atmosphere.

**Vector production and purification**. The murine *Asl* (*mAsl*) gene (Vega Sanger Asl-0003 transcript OTTMUST00000085369) inserted in a pCMV-SPORT6 gateway vector was purchased from Thermo Scientific (Loughborough, UK). A single-stranded AAV2 backbone plasmid containing an expression cassette with the EFS promoter, a modified simian virus 40 (SV40) small t antigen intron, the human vacuolar protein sorting 33 homologue (*hVPS33B*) cDNA, Woodchuck hepatitis post regulatory element (WPRE) sequence, SV40 late polyA (courtesy of P. Gissen) was digested with EcoRV-Nhe1 to remove the *hVPS33B* cDNA. Subsequently, the *mAsl* cDNA was digested with EcoRV-Nhe1 and ligated into this vector. Vector

production was performed by triple transfection in HEK293T cells following polyethylenimine transduction as described previously[60]. Vector purification was performed by affinity chromatography on an ÄKTAprime plus (GE Healthcare UK Ltd, Buckinghamshire, UK) with Primeview 5.0 software with a HiTrap AVB Sepharose column (GE Healthcare UK Ltd, Buckinghamshire, UK) according to the manufacturer's instructions. Vector quantification was performed by electro-phoresis on an alkaline gel[61]. An AAV8 vector encapsidating a single-stranded DNA sequence containing the *GFP* gene under the transcription activity of the EFS promoter, the SV40 intron upstream and WPRE and polyA downstream *GFP* was generated. The vector genomes tested contained 40 bp (TGTAGTTAATGATTA ACCCGCCATGCTACTTATCTACGTA) downstream of the 5′ITR and 45 bp (ATGGCTACGTAGATAAGTAGCATGGCGGGTTAATCATTAACTACA) just proximal of 3′ITR from the liver-specific enhancer-promoter element identified by Logan et al. in the wild-type AAV2 genome[62].

**Stereoscopic fluorescence microscopy**. At 5 weeks of age, CD-1 mice injected intravenously at day 0 with $1.7 \times 10^{11}$ vg per mouse of AAV8.EFS.*GFP* and control littermates were culled by terminal exsanguination and perfused with PBS. GFP expression was assessed using a stereoscopic fluorescence microscope (MZ16F; Leica, Wetzlar, Germany). Representative images were captured with a microscope camera (DFC420; Leica Microsystems, Milton Keynes, UK) and software (Image Analysis; Leica Microsystems).

**Free-floating and paraffin-embedded immunohistochemistry**. Brain tissue was fixed in 4% paraformaldehyde (PFA) over 48 h then stored in 30% sucrose at 4 °C. Cryo-sectioning was performed with a Microm freezing microtome (Carl Zeiss, Welwyn Garden City, UK). Immunohistochemistry was performed on free-floating sections blocked in Tris-buffered saline–Triton (TBST)/15% normal goat serum and incubated overnight at 4 °C using primary antibodies diluted in TBST/10% normal goat serum. After three washes with Tris-buffered saline (TBS), a 2 h incubation with a biotinylated secondary antibody at room temperature was followed by three TBS washes before a 2-h incubation with avidin-biotinylated horseradish peroxidase (ABC; 1:100; Vector, Peterborough, UK) at room temperature. After three TBS washes, detection was performed with a 0.05% 3,3′-diaminobenzidine (DAB) solution diluted in TBS. The reaction was stopped by the addition of ice-cold TBS. Three TBS washes were performed before mounting the tissue on chrome-gelatin-coated slides. Slides were cover-slipped with DPX-new (Merck Millipore Corporation, Temecula, CA, USA).

Systemic organs were fixed with 10% formalin for 48 h and stored in 70% ethanol at 4 °C. Paraffin-embedded sections were dewaxed, dehydrated in an ethanol gradient. Blocking was performed with 1% hydrogen peroxide in methanol for 30 min followed by antigen retrieval using 10 mmol l$^{-1}$ sodium citrate buffer pH 7.4. Nonspecific binding was blocked with 15% normal goat serum (Vector, Burlingame, CA, USA). After three washes in PBS, sections were incubated with primary antibodies overnight at 4 °C. Detection was performed with Polink-2 HRP Plus Rabbit Detection System for Immunohistochemistry (GBI labs, Mukilteo, WA, USA) as per the manufacturer's instructions. After dehydration in a gradient of ethanol and three washes in xylene, slices were cover-slipped with DPX-new (Merck Millipore Corporation, Temecula, CA, USA). Images were captured with a microscope camera (DFC420; Leica Microsystems, Milton Keynes, UK) and software (Image Analysis; Leica Microsystems).

**Free-floating immunofluorescence**. Free-floating sections were blocked in TBST/15% normal goat serum and incubated overnight at 4 °C with primary antibodies diluted in TBST/10% normal goat serum as described previously[41]. After three washes with TBS, samples were incubated for 2 h with secondary antibodies diluted in TBST/10% normal goat serum. After a further three washes, sections were incubated with DAPI (1:2000; Invitrogen) and mounted on chrome-gelatin-coated slides and cover-slipped with Fluoromount (Southern Biotech, Birmingham, AL, USA). For NOS isoforms and nitrotyrosine, blocking and incubation of antibodies were performed with 2% casein in TBST only.

All immunofluorescence images were acquired using an inverted Leica TCS SPE3 confocal microscope using ×20 (multi immersion with numerical aperture (NA) 0.6) and ×63 (oil immersion with NA 1.3) objectives and a 1.5 optical zoom for both objectives. The pinhole was set to one Airy Unit. The scan format was set at 1024 × 1024 pixels. Leica Application Suite Advanced Fluorescence software was used for basic analysis of the confocal images. Fiji software (ImageJ 1.50d) was used to project the z-stacks made on the ×63 objective in 2D using the Fiji tool: Image > Stacks > Z projection[63]. Both types of projection: Maximum intensity or Sum slices were used depending on the background level of each stack. Representative images are shown in all experiments. Colocalisation was performed using the Fiji plugin JACoP [626] and was represented by Pearson's coefficient calculated on images at ×20 objectives, after Costes randomisation and automatic threshold calculation[64].

**TUNEL staining**. TUNEL staining was performed as described previously[65] using the Roche kit (Roche, Welwyn Garden City, Hertfordshire, UK). Briefly, sections were incubated in 3% hydrogen peroxide in methanol for 15 min and washed in 0.1 M phosphate buffer (PB) before incubation with terminal deoxytransferase (TdT) and deoxyuridine trisphosphate (dUTP) in a solution of 0.1% TdT, 0.15%

dUTP, 1% cacodylate buffer at 37 °C for 2 h. The reaction was stopped by incubating the section in TUNEL stop solution (300 mM NaCl, 300 mM sodium citrate) for 10 min. Sections were then washed in $3 \times 0.1$ M PB solution, incubated with avidin-biotinylated horseradish peroxidase (ABC; 1:100; Vector, Peterborough, UK) at room temperature for 1 h, washed four times in 10 mM PB and visualised with DAB enhanced with cobalt nickel. The reaction was stopped in 10 mM PB and washed twice in double-distilled (ddH$_2$O) water.

**Nissl staining.** Brain sections were fixed in 4% PFA for 24 h then in 70% ethanol for 24 h. On day 3, sections were incubated in Cresyl Violet solution (BDH, East Grinstead, West Sussex, UK) for 3 min followed by dehydration in an ethanol gradient (70%, 90%, 96%, 96% with acetic acid, 100%), isopropanol, and three washes in xylene before being cover-slipped with DPX-new (Merck Millipore Corporation, Temecula, CA, USA).

**Quantitative analysis of immunological staining.** Ten random images per sample were captured with a microscope camera (DFC420; Leica Microsystems, Milton Keynes, UK) and software (Image Analysis; Leica Microsystems). Quantitative analysis was performed with thresholding analysis using the Image-Pro Premier 9.1 software (Rockville, MD, USA).

**Blood chemistry.** Plasma ammonia and alanine aminotransferase (ALAT) were analysed by Chemical Pathology Great Ormond Street Hospital, London.

**Mass spectrometry.** Blood was spotted onto a Guthrie card and allowed to dry at room temperature for 24 h. Amino acids were measured in dried blood spots by liquid chromatography-tandem mass spectrometry (LC-MS/MS). A 3-mm-diameter punch was incubated for 15 min in a sonicating water bath in 100 μL of methanol containing stable isotopes used as internal standards (2 nmol l$^{-1}$ of L-Arginine-13C; CK isotopes, Ibstock, UK) and L-Citrulline-d7 (CDN istotopes, Pointe-Claire, Quebec, Canada). A 4:1 volume of methanol was added to precipitate contaminating proteins. The supernatant was collected and centrifuged at $16,000 \times g$ for 5 min. Amino acids were separated on a Waters Alliance 2795 LC system (Waters, Midford, USA) using a XTerra® RP18, 5 μm, $3.9 \times 150$ mm column (Waters, Midford, USA). The mobile phases were (A) methanol and (B) 3.7% acetic acid. The gradient profile using a constant flow rate of 0.2 mL min$^{-1}$, with initially 100% B for the first minute and gradually increasing the flow of A as follows: 85% from 1 to 6 min, 75% from 6 to 8 min, 5% from 9 to 15 min, 100% from 16 to 25 min. The column was reconditioned for 10 min at the end of each run. Detection was performed using a tandem mass spectrometer Micro Quattro instrument (Micromass UK Ltd, Cheshire, UK) using multiple reaction monitoring in positive ion mode and ion transitions published previously[66]. The temperature of the source and for desolvation were 120 and 350 °C, respectively. The capillary and cone voltages were 3.7 and 35 V, respectively. The cone gas flow was 50 L h$^{-1}$ and the syringe pump flow 30 μL min$^{-1}$. The mass spectrometer vacuum was $4.3 \times 10^{-3}$ mbar. The multiplier and extractor voltages were 650 and 1 V, respectively. Data were analysed using Masslynx 4.1 software (Micromass UK Ltd, Cheshire, UK). Calibration curves ranging from 0 to 500 μM were constructed to enable quantification.

**Analysis of ASL enzyme activity.** Liver and brain samples were snap-frozen in dry ice at time of collection after perfusion of the animal with PBS to remove residual blood in tissues. Protein extraction was performed on ice. Samples were homogenised in lysis buffer (50 mM Tris, 150 mM NaCl, 1% Triton adjusted to pH 7.5) and centrifuged at $16,000 \times g$ for 20 min at 4 °C. Protein quantification of the supernatant was performed using the Pierce$^{TM}$ BCA protein assay kit (Thermo-Fisher Scientific, Rockford, IL, USA) according to the manufacturer's instructions.

Liver ASL activity was measured in duplicate samples. 20 μg protein was added to a buffer solution of 20 mM Tris, 1 mM argininosuccinic acid, 0.02 nM L-citrulline-d7 and incubated for 2 h at 37 °C. Brain ASL activity was also measured in duplicate samples. 80 μg protein was added to a buffer solution of 20 mM Tris, 30 μM argininosuccinic acid, 0.02 nM L-citrulline-d7 and incubated for 2 h at 37 °C. A 4:1 volume of methanol was added to stop the reaction, and centrifuged at $9500 \times g$ for 2 min. The supernatant was analysed by the LC-MS/MS method described above. ASL activity was calculated by subtracting the amount of argininosuccinic acid postincubation from that preincubation.

**Nitrite and nitrate measurement.** Measurement of nitrite and nitrate levels were performed using a modified Griess reaction protocol[67]. Samples were collected carefully in order to minimise the risk of nitrite and nitrate contamination. All glassware and plastic ware were cleaned with double-distilled water (ddH$_2$O). Animals were anaesthetised and perfused on ice. Organs and brain were collected on ice in less than 3 min and snap-frozen on dry ice. Samples were homogenised in two volumes of ddH$_2$O with a grinder (Tissue Master 125, OMNI International, Kennesaw, GA, USA) on ice and then centrifuged at $13,500 \times g$ at room temperature for 10 min in 3000 kDa cut-off filters (Merck Millipore, Darmstadt, Germany). Enzyme stocks were made and stored as follows: Nitrate reductase was dissolved in 1:1 phosphate buffer:glycerol to a final specific activity of 59 units mL

$^{-1}$ and stored at −20 °C. Glucose-6-phosphate dehydrogenase was dissolved in 1:1 phosphate buffer:glycerol to a final specific activity of 125 units mL$^{-1}$ and stored at −20 °C. The enzyme master mix was made fresh for each experiment using a mixture of 1 mL of phosphate buffer, 10 μL of nitrate reductase stock solution, 10 μL of glucose-6-phosphate dehydrogenase stock solution and 1.14 mg of glucose-6-phosphate. A calibration curve, covering a range of 0–600 μmol L$^{-1}$ was prepared using serial dilutions of nitrite and nitrate standards. Analysis was performed in a 96-well plate and each sample analysed in duplicate. 60 μL of sample was mixed with 10 μL of 10 mmol L$^{-1}$ NADPH and 40 μL of enzyme mastermix. Samples were incubated for 1 h at room temperature on a rotating shaker to allow conversion of NO$_3^-$ to NO$_2^-$. The Griess reaction was performed by adding 75 μL of 116 mM sulfanilamide, 5% phosphoric acid, then 75 μL of 7.7 mM N-1-naphtylethylenediamine dihydrochloride. The plate was read at 550 nm in a FLUOstar Omega spectrophotometer (BMG Labtech, Ortenberg, Germany).

**Nitrosothiols measurement.** Brain samples from PBS perfused mice were flash-frozen in dry ice and stored at −80 °C. Nitrosothiols were measured using a Nitrosothiols kit (Enzo Life Sciences, Farmingdale, NY, USA) according to the manufacturer's instructions after filtration of samples through a 10,000 kDa cut-off filters (Merck Millipore, Darmstadt, Germany). Samples were read at 540 nm using a FLUOstar Omega spectrophotometer (BMG Labtech, Ortenberg, Germany).

**cGMP measurement.** Brain samples from PBS perfused mice were flash-frozen in dry ice and ground in liquid nitrogen. Samples were weighed and diluted in ten volumes of 0.1 M HCl prior to centrifugation at $600 \times g$ for 10 min. cGMP was measured using a cGMP complete ELISA kit (Enzo Life Sciences, Farmingdale, NY, USA) according to the manufacturer's instructions in a nonacetylated format reaction. Samples were analysed in a 96-well plate and were analysed at 405 nm in a FLUOstar Omega spectrophotometer (BMG Labtech, Ortenberg, Germany).

**Glutathione analysis.** Reduced glutathione was measured as described previously[68]. Briefly snap-frozen samples were homogenised in 20 volumes of cold 50 mM Tris buffer pH 7.4 and sonicated. Monochlorobimane (at a final concentration of 100 μM) and glutathione-S-transferase (1 U mL$^{-1}$) were added to the samples prior to incubation at room temperature while protected from light for 30 min. Samples were analysed in a FLUOstar Omega spectrophotometer (BMG Labtech, Ortenberg, Germany) using an excitation wavelength of 360 nm and an emission wavelength of 450 nm.

**Green fluorescent protein ELISA.** Lysis buffer was added to cover the sample prior to grinding with a mixer after a couple of freeze/thaw cycles. The sample was then centrifuged for 2 min at $1200 \times g$ and the supernatant collected. The protein concentration for each sample was measured using the Pierce$^{TM}$ BCA protein assay kit (ThermoFisher Scientific, Rockford, IL, USA) according to the manufacturer's instructions and read at 570 nm in a FLUOstar Omega spectrophotometer (BMG Labtech, Ortenberg, Germany). Each step of the ELISA protocol was separated by three washes of a wash buffer 0.05% Tween20 in PBS. A monoclonal anti-GFP antibody (1:10,000; ab1218, Abcam, Cambridge, UK) was added, the plate sealed, incubated overnight at 4 °C then blocked with 1% bovine serum albumin in PBS for 1 h at 37 °C. Analysis was done in a sealed 96-well plate and all samples were measured in duplicate. GFP standards were serially diluted in wash buffer and incubated alongside a buffer blank and the samples for 1 h at 37 °C. An anti-GFP biotin-conjugated secondary antibody (1:5000; Ab6658, Abcam, Cambridge, UK) followed by a streptavidin-horseradish peroxidase conjugate (1:20,000; SNN2004, Invitrogen, Camarillo, CA, USA) were added to the samples. Both were incubated for 1 h at 37 °C successively. Tetramethylbenzidine was then added for 2 min at room temperature before the reaction was stopped with 2.5 M H$_2$SO$_4$. The plate was read at 450 nm within 30 min of having stopped the reaction in a FLUOstar Omega spectrophotometer (BMG Labtech, Ortenberg, Germany).

**qPCR.** Liver samples were stored frozen at −80 °C before DNA extraction with the DNeasy blood and tissue kit (QIAgen, Crawley, UK) according to the manufacturer's instructions. The WPRE sequence was amplified using the following primers: 5′-TTCCGGGACTTTCGCTTTCC-3′ (sense) and 5′-CGACAACACC ACGGAATTG-3′ (antisense). Amplification was detected and normalised against glyceraldehyde 3-phosphate dehydrogenase which was amplified using the following primers: 5′-ACGGCAAATTCAACGGCAC-3′ (sense) and 5′-TAGTGG GGTCTCGCTCCTGG-3′ (antisense). Amplification reactions were carried out using 5 μL of sample, 2.5 μmol L$^{-1}$ of each primer, and SYBR green master mix using the Quantitect SYBR Green PCR Kit (QIAgen, Crawley, UK) for a 25 μL reaction. The amplification conditions were 95 °C for 10 min followed by 40 cycles of 95 °C for 15 s, 60 °C for 1 min, 72 °C for 30 s. Data were processed with StepOne$^{TM}$ software v 2.3 (ThermoFisher Scientific, Rockford, IL, USA).

**Western blot.** Brain samples from PBS perfused mice were flash-frozen in dry ice and stored at −80 °C. Protein extraction was performed on ice. Samples were homogenised in Pierce$^{TM}$ RIPA buffer (ThermoFisher Scientific, Rockford, IL, USA) containing cOmplete$^{TM}$, Mini, EDTA-free Protease Inhibitor Cocktail

(Roche, Welwyn Garden City, Hertfordshire, UK) and incubated with gentle rotation at 4 °C for 30 min. After sonication, the lysate was centrifuged at 14,000 × $g$ for 20 min at 4 °C. Protein quantification of the supernatant was performed using the Pierce™ BCA protein assay kit (ThermoFisher Scientific, Rockford, IL, USA) according to the manufacturer's instructions. Supernatants were mixed with 5× SDS loading buffer and boiled for 5 min. Samples were then analysed by western blotting as previously published[69]. Uncropped membranes are presented in Supplementary Fig. 13.

**Statistics**. Data were analysed using GraphPad Prism 5.0 software (San Diego, CA, USA). Comparisons of continuous variables between two or more experimental groups were performed using the Student's two-tailed $t$ test or one-way ANOVA with Dunnett's post-test for pairwise comparisons with WT or untreated $Asl^{Neo/Neo}$ mice as indicated. $p$ values < 0.05 were considered statistically significant. For nonnormally distributed data, a log transformation was used to compare groups. Figures show mean ± standard error of the mean (SEM). Kaplan−Meier survival curves were compared with the log-rank test. Retrospective power calculation was performed by PS: Power and Sample Size Calculation programme version 3.1.2, 2014[70].

**Data availability**. The authors declare that all data supporting the findings of this study are available within the article and its supplementary information files or are available from the corresponding author upon request.

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

## Acknowledgements

We thank Mr. I. Doykov, Dr. F. Mazzacuva, Dr. E. Reid, and Mr. M. Wilson for technical assistance with LC-MS/MS; Mr. B. Warburton, Mrs. G. Sturges and Mrs. S. Richards for assistance with animal work; Dr. L. Lisowski (Children's Medical Research Institute, University of Sydney Westmead, Australia) for assistance in precisely identifying promoter-enhancer activity within the ITRs of the vector constructs. This work was supported by Action Medical Research for Children Charity (grant GN2137). P.B.M. is in receipt of a Great Ormond Street Hospital (GOSH) Children's Charity Leadership award (V2516). P.B.M. and P.G. are supported by the NIHR Great Ormond Street Hospital Biomedical Research Centre (the views expressed are those of the author(s) and not necessarily those of the NHS, the NIHR or the Department of Health). P.G. is a senior Welcome Trust fellow. S.M.B. and S.N.W. received funding from ERC grant "Somabio" (260862). S.N.W. received funding from MRC NC3Rs grant (NC/L001780/1). R.K. was part-funded by Borne charity. A.A.R. and M.P.H. receive funding from the UK Medical Research Council (MR/N026101/1). A.A.R. is also funded by the EU Horizon 2020 (BATCure, 666918). J.A.D. is supported by CONICYT Becas Chile Doctoral Fellowship programme 72160294.

## Author contributions

J.B., S.H., S.J.H., S.M.B., P.B.M., P.G., S.N.W. contributed to overall study design. J.B. performed experiments, contributed to the overall study design and wrote the manuscript, which was edited by all co-authors. D.P.P., J.H., M.L., E.R.-F., J.N., R.K., N.S., M.H., M.P.H., B.B., A.A.R., A.V., J.A.D. provided substantial assistance in experiments. H.P. analysed plasma samples. D.A.R. provided statistical assistance.

## Additional information

**Competing interests:** The authors declare no competing interests.

