## [Peer Review File · Nature Communications]

Reviewers' comments:

Reviewer #1 (Remarks to the Author):

Baruteau et al describe the results of a successful gene transfer study to treat a devastating urea cycle disorder, argininosuccinic aciduria, in mice. They describe two vectors in this report : one using the EFS promoter driving the expression of a GFP reporter, and another configured to express the Asl gene. The Asl mouse model has been previously described and is available from Jackson Labs in the USA. After establishing that their ssAAV8 vector resulted in hepatoneuronal expression of GFP, the Asl mutant animals were treated with the therapeutic vector. The results were quite striking in that the Asl neonates were rescued and had reduced evidence of corrected cerebral NO whilst the adults were not corrected. Conversely, Asl adults had robust correction of the urea cycle after AAV gene therapy as adults but not the animals treated as neonates. A number of behavioural, enzymatic, and histological studies accompany the description of the rescue. In the end, and based on the work herein, the authors propose an extension of the results to humans by treating with an AAV that will correct the liver and neurons, and then a second AAV that corrects the liver. This is a novel approach to the treatment of ASL in humans, a disorder that lacks effective therapies other than transplantation and , as the authors note, has CNS complications. The paper is well written and interesting to read, especially the discussion. These studies are an important extension of the previously successful liver directed gene therapy reported by Dr Lee's group (Baylor) where the Asl mice were quite convincingly rescued with liver directed gene therapy using an hAd vector (reference 21). The current work harnesses a unique property of some AAV serotypes to transduce the BBB and hence correct neurons and hepatocytes. The concept of using systemic AAV to treat hepatocerebral disease is well established in many studies using AAVs to treat mice with storage, systemic and neurodegenerative disorders. Whether an approach as suggested by the authors using a serotype 8 AAV will ever work in humans is questionable. The preferred serotypes for accessing the CNS after systemic delivery are AAV9, possibly AAV rh10, and more recently described novel serotypes, such as AAV PHP.B. The latter serotype (Deverman BE, et al. Cre-dependent selection yields AAV variants for widespread gene transfer to the adult brain. Nat Biotechnol. 2016 Feb;34(2):204-9. PMID: 26829320.) would be quite useful to test in the adult Asl mice to address the neonatal vs adult differences because this serotype is claimed to cross the BBB about 40x greater than AAV9, the standard (reference 45), in adults.

Another set of studies that would strengthen the paper would be to generate and test a series of AAV vectors with neuronal vs liver promoters, and use serotype(s) such as AAV9 or PHP.B to dissect which cell(s) must be corrected and when for benefit. The essential element of the cassette is made and tested, and such studies might be strongly considered as they would bolster and extend the significance of a second report of successful gene therapy in Asl mice. One fascinating question would be to correct the brain only using an appropriate AAV, support the urea cycle with acylation and arg, and then withdraw the diet and assess the phenotype. Similarly, the use of an PHP.B vector might allow adult brain correction to be studied after medical rescue.

Detailed remarks to improve paper:

1. Line 57. Extreme caution should be used to describe conditional neuronal targeting. This section should be reworded.
2. The neuronal correction of the neonates is present but minor. Where else are neurons transduced? More sections and staining would help.
3. The AAV doses are reported as vg/mouse. On line 183, 3×10^{11} vg are being given to neonates; this equates to doses greater than have ever been given to humans, to date per this reviewers knowledge. How translatable might the vector route and dose be? What is the lowest effective dose? Comments and experiments are needed along these lines.
4. Figure 2- some might be moved to supplemental and expanded to focus more on the brain – regions and cell types.
5. Figure 3 – there are small numbers of mice used – more animals should be generated or stated why this might not be feasible. The legend states 5 mice for some groups.
6. Figure 4 – might be compressed or moved to supplemental.

7. Figure 7 – very interesting but not fully enabled by the data in this manuscript.

Reviewer #2 (Remarks to the Author):

Argininosuccinic aciduria (ASA) is the second most common urea cycle disorder, which is often characterized by hyperammonaemia and other late-onset phenotypes, including neurocognitive impairment. In this manuscript, using a hypomorphic ASLNeo/Neo mouse model, the authors first characterized the neuropathophysiology of ASA. They found that downstream metabolites of nitric oxide (NO) and cGMP are increased in brain homogenates from ASLNeo/Neo mice. They also observed increased immunostaining against nitrotyrosine in neurons from ASLNeo/Neo mice and concluded that nitrosative stress plays a role in the neuropathy of ASA. The authors then performed AAV8-mediated gene therapy to test whether they can rescue the phenotypes in ASLNeo/Neo mice. They found that AAV8 injection in the adult – but not neonatal - mice provides long-term correction of the urea cycle and the macroscopic phenotypes. Interestingly, long term correction of cerebral NO metabolism could only be achieved when gene therapy were performed neonatally. Neonatal injection of AAV8 gene therapy resulted in behavioural improvement and decreased cortical apoptosis.

The major novelty presented in the manuscript includes the very first characterization of brain phenotypes in mouse model of ASA. The authors also performed the first proof-of-concept study of AAV-mediated gene therapy in ASA. Gene therapy in ASA is not new as it had been previously shown using adenovirus though overall survival has been reported, the neurological phenotype was not described. Hence the gene therapy study presented in this manuscript potentially into the neurological phenotype in ASA. However, the study suffers from several major design flaws that limit the conclusions that can be drawn from this work including biochemical endpoints of NO function, quantification of biochemical endpoints, and choice of control animals.

Major Comments.

1. The authors claimed that there's an increased formation of NO as assessed by nitrite/nitrate levels. This seems to contradict the previous findings that loss of ASL leads to systemic reduction of NO. Erez et al previously found that nitrite concentration in brain isn't changed in ASLNeo/Neo mice. However, they observed reduced RSNO level in the brain of ASLNeo/Neo mice (even though it's not statistically significant). Nitrate/Nitrite level from brain homogenates might not truly represent physiological level of nitric oxide and thus, the authors need to show other ways of measuring nitric oxide concentration, such as RSNO level measurement. Furthermore, Nagamani et al showed that NO supplementation was associated with improvement in cognitive measures in a subject with ASA. Hence, if the data shows consistent increase of NO, I think the authors need to elaborate more in the discussion section why.
2. The authors assess nitrite and nitrate in wild type and Neo/Neo mice. However, the age of these mice is unclear. Moreover, in the methods, it states that only the Neo/Neo mice were treated with a low protein diet, sodium benzoate and arginine whereas wild type mice were treated with standard chow. If wild type mice are not similarly treated, it is difficult to draw conclusions from their results especially since the dietary nitrite/nitrate content might be different in the 2 chows. Moreover, the use of different diet in Neo/Neo mice vs. wild type and heterozygous mice suggests that the mice were not housed together which might also introduce environmental variation. The authors should be more clear on how the mice are housed and why all mice are not treated with the same diet and medications. Also, addition of the ages of mice is important as the interval between these differing treatments and the studies remains unclear in gene therapy treated mice.
3. If the nitrosative stress is caused by low arginine-mediated NOS uncoupling, can arginine treatment rescue the brain phenotypes? The authors need to address this.
4. The authors performed immunostaining to prove that there's increased nitrosative stress in the brain of ASLNeo/Neo mice. Quantification of nitrotyrosine immunostaining needs to be provided in every figure. Furthermore, western blot using anti-nitrotyrosine on brain homogenates would

probably be a more convincing way to show that there is an increase in nitrosative stress.

5. For figure 1, western blot data on GFAP, CD68, nNOS, iNOS, and eNOS needs to be shown especially since Erez et al. has previously shown that eNOS and nNOS protein level remains unchanged in the brain homogenates of ASLNeo/Neo mice.

6. The authors conclude that there is reduced glutathione in the brain but that results fell short of significance. However, in Figure 1c, there seems to be wide variation in results with two samples similar to WT and two samples much lower. The authors should comment on this wide variation and consider analysis with larger sample sizes to better elucidate this point.

7. On Figure 1(i), how do the authors identify endothelial cells? Morphologically? Double-staining with endothelial markers, such as CD31 or CD144, is needed to specifically prove that eNOS expression is increased in brain endothelial cells.

8. The authors noted differences in behavioral studies such as rotarod. However, these mice are different sizes (see Figure 3). Some behavioral studies are confounded by size differences and should not be used to make definitive conclusions.

9. Please indicate the age when cerebral NO metabolism is assessed in gene therapy-treated mice (Figure 5)

10. In Figure 6, the authors evaluate apoptotic cells in the cortex. However, the age range of the untreated mice is 21 days to 12 months whereas the other groups are 9-12 months of age. It is unclear whether one can compare the brain of a 21 day old mouse with a mouse of 9-12 months of age. It is unclear why such a large age range in untreated mice was used.

11. The authors showed that nitrite/nitrate levels are normalized in neonatally injected mice. Again, this is not a sufficient marker. What about cGMP level? Protein nitrosylation?

12. Reduced liver glutathione was shown to be decreased in untreated ASLNeo/Neo mice (Supplementary Figure 8B). Was this changed in gene therapy-treated mice?

13. It is unclear why the authors did not use an "empty" virus control in these experiments.

Minor comments:

1. The graphic of the urea cycle in Supplementary Figure 1 is confusing. For instance, citrulline is not included in the "urea cycle" portion of the graphic although citrulline is an intermediate in the urea cycle.

2. On page 16, the authors state that hypoargininemia likely causes the hair phenotype. However, if this is the case, they do not speculate as to why this hair phenotype is unique to argininosuccinic aciduria as compared to other urea cycle disorders which are also associated with low arginine levels.

3. In Figure 6(c), the line and the 'ns' sign comparing the statistical difference between WT and neonatally -injected mutants are partly missing.

4. In Supplementary Figure 2, please indicate the age of mice when the brain tissues were collected.

5. Figure 3: "(d, e) Mean growth of neonatally injected ASLNeo/Neo mice compared to WT (n=31) and untreated ASLNeo/Neo mice (n=41) (D) during the first month (n=13) and (e) over 9 months (n=7)." Letter (D) needs to be in lower case.

6. The authors should fix the numerous typographical errors throughout the manuscript.

Reviewer #3 (Remarks to the Author):

The manuscript by Baruteau et al., describes that nitrosative/oxidative stress is increased in ASL-deficient mice, and that AAV-mediated expression of ASL in mice reverses pathological impairments caused by ASL deficiency. The former (i.e., nitrosative stress in the brain) is a novel finding, and the results are mostly convincing. However, the report overall is not informative and will only have minimal contribution to the field due to the reasons listed below. Thus, I feel that the report is not appropriate for Nature Communications.

1. Most experiments reported in this report show that AAV-mediated restoration of ASL expression

in ASL deficient mice can correct the pathological phenotypes seen in the mice. These are completely expected results (since the authors are simply putting back the same gene that were removed from the mouse) and thus do not bring any new scientific information.

2. ASL deficiency leads to decreased NO generation in liver, since ASL is coupled to NO synthase. In contrast, the authors claim that NO production and the cGMP signaling are upregulated in the brains of ASL-deficient mice. However, why ASL deficiency leads increased NO synthesis in the brain is not explained in the manuscript. The authors speculate that limited availability of GSH in neurons can cause increased nitrotyrosine staining, but the limited GSH in the brain cannot explain why NO production (and cGMP formation) is increased in the brain.

3. It was very difficult for this reviewer to understand the experiment scheme for Figure 3 (animal injections and analysis) described in page 8. For instance, only 25% of untreated ASL KO mice survived after the injection; did the authors monitor the growth of survived animals for 12 month? Extensive editing is recommended for clarity.

4. In Figure 1, the authors need to conduct a co-staining experiment with a neuronal marker to prove that nitrotyrosine, nNOS, and iNOS are indeed increased in neurons.

5. Did "untreated" control Asl KO mice receive "vehicle" injections? Controls should be vehicle-injected animals.

Minor concerns

i) Figure citation is not in a Nature Communications format (for instance, Fig. 1A should be Fig 1a).

ii) In Fig. 2, EFS is not explained in the figure legend. I think it should be labeled "AAV" or "GFP" in this Figure.

iii) Figure 3a and 3b are not cited in the text.

Reviewers' comments:

Reviewer #1 (Remarks to the Author):

Baruteau et al describe the results of a successful gene transfer study to treat a devastating urea cycle disorder, argininosuccinic aciduria, in mice. They describe two vectors in this report : one using the EFS promoter driving the expression of a GFP reporter, and another configured to express the Asl gene. The Asl mouse model has been previously described and is available from Jackson Labs in the USA. After establishing that their ssAAV8 vector resulted in hepatoneuronal expression of GFP, the Asl mutant animals were treated with the therapeutic vector. The results were quite striking in that the Asl neonates were rescued and had reduced evidence of corrected cerebral NO whilst the adults were not corrected.

Conversely, Asl adults had robust correction of the urea cycle after AAV gene therapy as adults but not the animals treated as neonates. A number of behavioural, enzymatic, and histological studies accompany the description of the rescue. In the end, and based on the work herein, the authors propose an extension of the results to humans by treating with an AAV that will correct the liver and neurons, and then a second AAV that corrects the liver. This is a novel approach to the treatment of ASL in humans, a disorder that lacks effective therapies other than transplantation and , as the authors note, has CNS complications. The paper is well written and interesting to read, especially the discussion.

These studies are an important extension of the previously successful liver directed gene therapy reported by Dr Lee's group (Baylor) where the Asl mice were quite convincingly rescued with liver directed gene therapy using an hAd vector (reference 21). The current work harnesses a unique property of some AAV serotypes to transduce the BBB and hence correct neurons and hepatocytes. The concept of using systemic AAV to treat hepatocerebral disease is well established in many studies using AAVs to treat mice with storage, systemic and neurodegenerative disorders. Whether an approach as suggested by the authors using a serotype 8 AAV will ever work in humans is questionable. The preferred serotypes for accessing the CNS after systemic delivery are AAV9, possibly AAV rh10, and more recently described novel serotypes, such as AAV PHP.B. The latter serotype (Deverman BE, et al. Cre-dependent selection yields AAV variants for widespread gene transfer to the adult brain. Nat Biotechnol. 2016 Feb;34(2):204-9. PMID: 26829320.) would be quite useful to test in the adult Asl mice to address the neonatal vs adult differences because this serotype is claimed to cross the BBB about 40x greater than AAV9, the standard (reference 45), in adults.

Another set of studies that would strengthen the paper would be to generate and test a series of AAV vectors with neuronal vs liver promoters, and use serotype(s) such as AAV9 or PHP.B to dissect which cell(s) must be corrected and when for benefit. The essential element of the cassette is made and tested, and such studies might be strongly considered as they would bolster and extend the significance of a second report of successful gene therapy in Asl mice. One fascinating question would be to correct the brain only using an appropriate AAV, support the urea cycle with acylation and arg, and then withdraw the diet and assess the phenotype. Similarly, the use of an PHP.B vector might allow adult brain correction to be studied after medical rescue.

We thank the reviewer for these comments.

We agree the vector might require some optimisation before clinical translation, especially with regards to the transgene cassette. The murine *Asl* gene will need to be replaced by a human *ASL* gene; codon-optimisation and different enhancers might increase the expression of the transgenic protein.

The suggestion of testing alternative capsids with better CNS transduction is debatable. Although the use of the AAV8 capsid has been restricted to liver-directed clinical trials so far, unpublished data of our group have demonstrated a similar efficacy in CNS transduction of AAV8 and AAV9 capsids in murine models after systemic injections (**Annex 1**) as observed by others after either intravenous (Yang et al, Mol Ther. 2014 Jul;22(7):1299-309. doi: 10.1038/mt.2014.68; Bish et al, Hum Gene Ther. 2008 Dec;19(12):1359-68. doi: 10.1089/hum.2008.123; Zincarelli et al, Mol Ther. 2008 Jun;16(6):1073-80. doi: 10.1038/mt.2008.76) or intracranial (Chakrabarty et al, PLoS One. 2013 Jun 25;8(6):e67680. doi: 10.1371/journal.pone.0067680; Aschauer et al, PLoS One. 2013 Sep 27;8(9):e76310. doi: 10.1371/journal.pone.0076310) injections.

AAV PHP.B is unlikely to be a suitable capsid for clinical translation as the liver transduction is poor (3-log lower compared to AAV9 in Deverman et al, Nat Biotechnol. 2016 Feb;34(2):204-9. doi: 10.1038/nbt.3440); this capsid displays great promises for CNS transduction with a much better efficacy compared to AAV9 and minimising off-target biodistribution but this is not appropriate in this disease model as both liver and CNS need to be efficiently transduced to correct the urea cycle and the cerebral disease, respectively. Moreover, AAV-PHP.B has not been proven to work in other species (Matsuzaki Y et al, Neurosci Lett. 2017 Nov 24;665:182-188. doi: 10.1016/j.neulet.2017.11.049). AAV capsids are known to have different transduction profile from species to species. Moreover, there are some preliminary conference reports, which indicate that AAV-PHP.B displays excellent CNS transduction in C57BL/6 mice but this is not observed neither in other mouse strains (BALB/cJ and CB6F1/J mice) nor non-human primates. An assumption is that the *in vivo* selection process described by Deverman et al, 2016 in their paper might explain this observation.

The suggestion of testing alternative promoters is theoretically of interest. However, identifying an exclusive neuronal promoter without a confounding liver off-target effect in this disease model is challenging. For example, experiments in our group have shown that the neuron-specific promoter, human synapsin hSyn, driving the expression of GFP in an AAV9 capsid injected intracranially in neonatal pups led to a widespread biodistribution in peripheral organs (Ng et al, Human Gene Ther, 2015; 26, A25) although this promoter is well identified as neuron-specific (Nathanson, Neuroscience. 2009 Jun 30;161(2):441-50. doi: 10.1016/j.neuroscience.2009.03.032; Jackson et al, Front Mol Neurosci. 2016 Dec 22;9:154. doi: 10.3389/fnmol.2016.00154). Most of the promoters used for transducing the brain in animal experiments or clinical trials are associated with central and peripheral transduction (CMV, PGK, U1a, CAG) (Hocquemiller, Hum Gene Ther. 2016 Jul;27(7):478-96. doi: 10.1089/hum.2016.087). Most of the claimed neuron-specific promoters cause as well liver transduction, as the biodistribution is not systematically either mentioned or investigated. As in the hypomorphic *Asf^{Neo/Neo}* mouse model, a very small percentage of transduced hepatocytes can normalise ammoniaemia. Therefore we think that investigating an exclusively neuronal-specific promoter is a very challenging experiment to perform in vivo as the low rate of hepatocyte transduction observed after intracranial injection might normalise ammonia levels and prevent robust conclusions on a model with restricted neuronal correction of ASL deficiency.

Comparing the EFS to a liver-specific promoter will practically repeat the experiments we described in injecting adult (model with correction of ammonia levels through liver transduction) and neonatal (model of with correction of ammonia levels through liver transduction and neuronal ASL correction through brain transduction) mice, taking advantage of the limited permeability of the blood brain barrier in adult mice.

Detailed remarks to improve paper:

1. Line 57. Extreme caution should be used to describe conditional neuronal targeting. This section should be reworded.

We thank the reviewer for this comment. This section has been reworded as suggested. Co-immunofluorescence of brain section has shown that the nitrosative stress identified by nitrotyrosine staining was involving neuronal cells (presented in amended Figure 1).

2. The neuronal correction of the neonates is present but minor. Where else are neurons transduced? More sections and staining would help.

We thank the reviewer for this comment. Neuronal correction after neonatal intravenous injection is predominant in cortical regions, and following a rostro-caudal gradient with gradually decreasing transduction, prominent in the forebrain, with no transduction in the cerebellum and brainstem (presented in amended Figures 2b and 2c).

3. The AAV doses are reported as vg/mouse. On line 183, 3×10^{11} vg are being given to neonates; this equates to doses greater than have ever been given to humans, to date per this reviewer's knowledge. How translatable might the vector route and dose be? What is the lowest effective dose? Comments and experiments are needed along these lines.

We thank the reviewer for this comment. 3×10^{11} vg in a pup weighing 1.5 gm corresponds to 2×10^{14} vg/kg. No mortality or adverse events were attributed to the vector itself. Further toxicity studies in large animal models will need to confirm these preliminary data.

Avexis, a gene therapy company, has recently reported a clinical trial in spinal muscular atrophy (AVXS-101, NCT02122952 in clinicaltrials.gov; analysis recently published in Mendell et al, *N Engl J Med*, 377(18):1713-1722. doi: 10.1056/NEJMoa1706198) with an AAV9 derived capsid with 2×10^{14} vg/kg in the high dose group (n=12 paediatric patients injected). Avexis is now recruiting for a phase 3 trial 15 additional patients at the high dose identified as a safe therapeutic dose (AVXS-101, NCT03306277 in clinicaltrials.gov).

4. Figure 2- some might be moved to supplemental and expanded to focus more on the brain – regions and cell types.

We thank the reviewer for this comment. The amended Figure 2 now focusses on brain transduction detailing brain regions and cell types targeted. The content related to liver transduction has been moved into Supplementary figure 5.

5. Figure 3 – there are small numbers of mice used – more animals should be generated or stated why this might not be feasible. The legend states 5 mice for some groups.

We thank the reviewer for this comment. More animals were injected but some were cannibalised by their dam. That's why these numbers are lower than initially planned. Most of these animals were cross-fostered with another strain to reduce the risk of maternal cannibalism. This has been added in the Material and Methods section.

Retrospective power calculation was performed by PS: Power and Sample Size Calculation programme version 3.1.2, 2014 (Dupont and Plummer, Controlled Clinical Trials, 1990). Power calculation based on liver ASL activity provides a power value of 1 (calculated with the following parameters: mean in untreated *Asl^{Neo/Neo}* mice 14.5%; mean in adult-treated *Asl^{Neo/Neo}* mice 47%; σ value (standard deviation in untreated *Asl^{Neo/Neo}* mice) 4%; sample size $n=5$; α value= 0.05). This means that if the true difference in the experimental and control means is 33, we will be able to reject the null hypothesis that the population means of the experimental and control groups are equal with probability (power) 1.000. The Type I error probability associated with this test of this null hypothesis is 0.05. This demonstrates that this sample size of 5 animals is sufficient to generate reliable conclusion. This calculation has been added in the Result section of the manuscript.

6. Figure 4 – might be compressed or moved to supplemental.

We thank the reviewer for this comment. This figure, which details the efficacy of gene therapy on the urea cycle, has been modified as suggested and panels describing ammoniaemia and plasma argininosuccinic acid levels at 2 months after injection and vector genome copies in liver samples were moved to Supplemental figure 9.

7. Figure 7 – very interesting but not fully enabled by the data in this manuscript.

We thank the reviewer for this comment. The legend acknowledges that all the pathways described are not fully demonstrated of the paper and mentions “Proposed pathophysiological mechanisms”. The picture provides a comprehensive summary of the data observed in the *Asl^{Neo/Neo}* mouse model in the research papers describing the pathophysiology of the disease and published over the last years (Erez et al 17(12):1619-26. doi: 10.1038/nm.2544; Nagamani, 90(5):836-46. doi: 10.1016/j.ajhg.2012.03.018; this study).

This figure relies on data provided in this study and in the literature:

- Low arginine and NOS uncoupling (Nagamani et al, Erez et al)
- Increase of ROS (plasma and urine isoprostane, superoxide ion in aorta in Nagamani et al)

- Nitrosative stress, increased cGMP and higher cell death rate in *Asl^{Neo/Neo}* mice compared to WT.
- Up-regulation of GSH synthesis in astrocytes but not neurons (Gegg et al. J Neurochem. 2003 Jul;86(1):228-37. PubMed PMID: 12807442).
- Hyperammonaemia alters the NH₃/Glutamate/NO/cGMP pathway via NMDA receptors (Braissant et al, J Inherit Metab Dis. 2013 Jul;36(4):595-612. doi: 10.1007/s10545-012-9546-2). Chronic hyperammonaemia increases guanylate cyclase activity in the cortex and decreases it in the cerebellum of patients and other animal models (Rodrigo et al, Neurochem Int. 2006 May-Jun;48(6-7):472-7. PubMed PMID: 16517021). This is in line with what we have observed with increase nitrite/nitrate levels in the cerebrum, the diencephalon/midbrain but not in the hindbrain of *Asl^{Neo/Neo}* mice (Supplementary figure 2).

These references discussed throughout the manuscript have been added in the legend of this figure for clarity.

Reviewer #2 (Remarks to the Author):

Argininosuccinic aciduria (ASA) is the second most common urea cycle disorder, which often characterized by hyperammonaemia and other late-onset phenotypes, including neurocognitive impairment. In this manuscript, using a hypomorphic ASLNeo/Neo mouse model, the authors first characterized the neuropathophysiology of ASA. They found that downstream metabolites of nitric oxide (NO) and cGMP are increased in brain homogenates from ASLNeo/Neo mice. They also observed increased immunostaining against nitrotyrosine in neurons from ASLNeo/Neo mice and concluded that nitrosative stress plays a role in the neuropathy of ASA. The authors then performed AAV8-mediated gene therapy to test whether they can rescue the phenotypes in ASLNeo/Neo mice. They found that AAV8 injection in the adult – but not neonatal - mice provides long-term correction of the urea cycle and the macroscopic phenotypes. Interestingly, long term correction of cerebral NO metabolism could only be achieved when gene therapy were performed neonatally. Neonatal injection of AAV8 gene therapy resulted in behavioural improvement and decreased cortical apoptosis.

The major novelty presented in the manuscript includes the very first characterization of brain phenotypes in mouse model of ASA. The authors also performed the first proof-of-concept

study of AAV-mediated gene therapy in ASA. Gene therapy in ASA is not new as it had been previously shown using adenovirus though overall survival has been reported, the neurological phenotype was not described. Hence the gene therapy study presented in this manuscript potentially into the neurological phenotype in ASA. However, the study suffers from several major design flaws that limit the conclusions that can be drawn from this work including biochemical endpoints of NO function, quantification of biochemical endpoints, and choice of control animals.

Major Comments.

1. The authors claimed that there's an increased formation of NO as assessed by nitrite/nitrate levels. This seems to contradict the previous findings that loss of ASL leads to systemic reduction of NO. Erez et al previously found that nitrite concentration in brain isn't changed in ASL^{Neo/Neo} mice. However, they observed reduced RSNO level in the brain of ASL^{Neo/Neo} mice (even though it's not statistically significant). Nitrate/Nitrite level from brain homogenates might not truly represent physiological level of nitric oxide and thus, the authors need to show other ways of measuring nitric oxide concentration, such as RSNO level measurement. Furthermore, Nagamani et al showed that NO supplementation was associated with improvement in cognitive measures in a subject with ASA. Hence, if the data shows consistent increase of NO, I think the authors need to elaborate more in the discussion section why.

We thank the reviewer for this comment. ASA is associated with systemic NO deficiency (Erez et al 17(12):1619-26. doi: 10.1038/nm.2544). However our study suggests that this finding doesn't apply to the brain where the pathophysiology is different. Erez et al found that nitrite concentration in a whole brain lysate is unchanged, which is not in contradiction with our data as this nitrate/nitrite increase was measured in cortical regions where the nitrotyrosine staining is the most prominent compared to other brain areas (See amended Figure 1e). This increase of nitrate/nitrite levels follows a decreasing rostro-caudal gradient as the nitrotyrosine staining in *Asl*^{Neo/Neo} mice is minimal in the cerebellum.

We assessed RSNO levels in brain homogenates and we observed a trend for increased levels in mutants compared to WT but without significance (Supplementary figure 2). These results are contradictory neither with increased nitrite levels in cortex nor results from Erez et al. In Erez et al and our experiment of nitrotyrosine, whole brain lysates were used. For experiment

assessing nitrite levels, cortical samples were used. Experiments using whole brain lysates can be misleading as the observed nitrosative/oxidative stress doesn't involve all brain areas and, within these areas, not to all neurons. In the authors' opinion, nitrotyrosine immunostaining is the best way to show it. Alternative quantification methods in whole brain lysates will not be meaningful if there is no difference and could potentially lead to erroneous conclusions as not taking into consideration enough the specificity and increased sensitivity of some brain regions. For example, we observed increased nitrite/nitrate levels in forebrain/midbrain but not in the hindbrain of *Asl^{Neo/Neo}* mice (Supplementary figure 2). Previous publications have highlighted opposite alterations of the NH₃/glutamate/NO/cGMP pathway during chronic hyperammonaemia with guanylate cyclase activity increased in the cortex and decreased in the cerebellum of patients and rodent animal models (Rodrigo et al, *Neurochem Int.* 2006 May-Jun;48(6-7):472-7. Epub 2006 Mar 6. Review. PubMed PMID: 16517021). This has been added in the discussion.

The discussion has been expanded to comment more on increased NO levels and the neurological effect of NO in one subject. Nagamani et al were unable to show IQ improvement and stated the results might have been biased by practice effect. Therefore the ongoing clinical trial (NCT 03064048) assessing the neurocognitive effect of NO supplementation in ASA patients will help in answering the question with more robust conclusions.

2. The authors assess nitrite and nitrate in wild type and Neo/Neo mice. However, the age of these mice is unclear. Moreover, in the methods, it states that only the Neo/Neo mice were treated with a low protein diet, sodium benzoate and arginine whereas wild type mice were treated with standard chow. If wild type mice are not similarly treated, it is difficult to draw conclusions from their results especially since the dietary nitrite/nitrate content might be different in the 2 chows. Moreover, the use of different diet in Neo/Neo mice vs. wild type and heterozygous mice suggests that the mice were not housed together which might also introduce environmental variation. The authors should be more clear on how the mice are housed and why all mice are not treated with the same diet and medications. Also, addition of the ages of mice is important as the interval between these differing treatments and the studies remains unclear in gene therapy treated mice.

We thank the reviewer for this comment and apologise for our lack of clarity and detail. Mice assessed for nitrite/nitrate in the brain were 2 to 4 month-old in Figure 1. This has been specified in the legend.

All WT and mutant littermates were housed together in the same cages and were under the same protein-restricted diet and water supplementation. The sentence was not clearly written in the Method section and has been reworded for clarity.

3. If the nitrosative stress is caused by low arginine-mediated NOS uncoupling, can arginine treatment rescue the brain phenotypes? The authors need to address this.

We thank the reviewer for this comment. ASL has a structural role to maintain a multiprotein complex, which includes NOS and a cationic transporter CAT1 for channelling arginine intracellularly. Mutated ASL protein will cause a loss of the integrity of this complex, which will prevent intracellular channelling of arginine (Erez et al, Nat Med. 2011 Nov 13;17(12):1619-26. doi: 10.1038/nm.2544). Therefore arginine supplementation alone cannot restore the NOS uncoupling caused by low arginine and correct the brain phenotype. This is what is observed in patients, who can sometimes have high arginine supplementation, normalised plasma arginine levels but without getting a better outcome (Baruteau et al, J Inherit Metab Dis. 2017 May;40(3):357-368. doi: 10.1007/s10545-017-0022-x).

4. The authors performed immunostaining to prove that there's increased nitrosative stress in the brain of ASLNeo/Neo mice. Quantification of nitrotyrosine immunostaining needs to be provided in every figure. Furthermore, western blot using anti-nitrotyrosine on brain homogenates would probably be a more convincing way to show that there is an increase in nitrosative stress.

We thank the reviewer for this comment. Quantification of nitrotyrosine staining is presented in Figures 1c (amended following reviewer's comment) and 5c.

Western blot using anti-nitrotyrosine on brain homogenates is presented in Supplementary Figure 2f. No significant difference in WB for nitrotyrosine in brain homogenates between WT and mutants was observed.

As discussed in Question 1 from Reviewer 2, brain homogenates provides a rough estimate of nitrotyrosine levels. This doesn't take into account specificities of brain areas and cell types. The increased nitrotyrosine staining was observed mostly in the cortex (Amended Figure 1e), involving neurons (Figure 1d) but not all of them (Figure 1c). Therefore a quantification of the nitrotyrosine staining in cortical slides provides a better assessment of the oxidative stress

in the cortex than an assay measuring this nitrotyrosine biomarker in the whole tissue lysate, which could lead to superficial or erroneous conclusions.

5. For figure 1, western blot data on GFAP, CD68, nNOS, iNOS, and eNOS needs to be shown especially since Erez et al. has previously shown that eNOS and nNOS protein level remains unchanged in the brain homogenates of ASLNeo/Neo mice.

We thank the reviewer for this comment. Quantification of the immunostaining for GFAP, CD68, nNOS, iNOS, and eNOS in cortical regions was performed using the Image Pro Premier software and is presented in novel Supplementary Figures 2g, 3a, 3c and 3e.

Similarly to the argumentation for RSNO and nitrotyrosine staining (Questions 1 and 4 of Reviewer 2, respectively), the increase of NOS isoforms is not observed in all neuronal (for nNOS and iNOS) and endothelial (for eNOS) cells. In Erez et al, western blots were performed from whole brain lysates, limiting the extrapolation of these conclusions to a limited population of neurons, mostly affected in cortical regions. Even though, an attempt to quantify the 3 NOS isoforms was performed but various troubleshooting occurred precluding to provide reliable results so far.

GFAP and CD68 immunostaining do not show any difference in staining in WT or *Asl^{Neo/Neo}* mice. Therefore western blot analysis has not been performed.

6. The authors conclude that there is reduced glutathione in the brain but that results fell short of significance. However, in Figure 1c, there seems to be wide variation in results with two samples similar to WT and two samples much lower. The authors should comment on this wide variation and consider analysis with larger sample sizes to better elucidate this point.

We thank the reviewer for this comment. This experiment has been repeated with a larger sample size. This confirmed the absence of significant difference of reduced glutathione levels in the cortex of WT or *Asl^{Neo/Neo}* mice. Results are presented in Supplementary Figure 2d. The retrospective power calculation was 0.371 when performed by PS: Power and Sample Size Calculation programme version 3.1.2, 2014 (Dupont and Plummer, Controlled Clinical Trials, 1990), including the following parameters: sample size of n=10 WT and n=9 *Asl^{Neo/Neo}* mice; means of WT and *Asl^{Neo/Neo}* mice of $11319 \pm 7119 \mu\text{M}$ and $17137 \mu\text{M}$; respectively; α value= 0.05. A power of 0.90 would require n=34 WT and n=31 *Asl^{Neo/Neo}* mice, which was not possible to achieve due to the lack of resources. This calculation has been added in the Result section of the manuscript.

The absence of difference cerebral reduced glutathione between WT and $Asl^{Neo/Neo}$ mice is not surprising as nitrosative/oxidative stress involves neuronal cells (Figure 1d) and decreased reduced glutathione levels in neurons might be compensated by the functional complementarity between neurons and astrocytes. As presented in the discussion and published previously (Gegg et al. J Neurochem. 2003 Jul;86(1):228-37. PubMed PMID: 12807442), “neurons are more vulnerable to oxidative stress than astrocytes *in vitro*, as they cannot overexpress γ -glutamyl transpeptidase to replenish their intracellular glutathione content. Therefore, they rely on the paracrine glutathione supply from astrocytes when exposed to reactive nitrogen species and oxidative stress.” This increased supply of reduced glutathione by astrocytes might mask the reduced neuronal levels. Tissue homogenate is unable to assess cell type specificities if cross-correction between cell types occur.

7. On Figure 1(i), how do the authors identify endothelial cells? Morphologically? Double-staining with endothelial markers, such as CD31 or CD144, is needed to specifically prove that eNOS expression is increased in brain endothelial cells.

We thank the reviewer for this comment. Colocalisation with eNOS and CD31 confirmed increased eNOS expression in endothelial cells (Supplementary Figure 3).

8. The authors noted differences in behavioral studies such as rotarod. However, these mice are different sizes (see Figure 3). Some behavioral studies are confounded by size differences and should not be used to make definitive conclusions.

We thank the reviewer for this comment. In the rotarod test, weight is known to influence the outcome. “Heavy mice are expected to perform worse than light mice” (Brooks, Nature Review Neuroscience; 2009, 10, 519-529) and this has been previously demonstrated (Kovacs et al, Sci Rep, 3, 2116; Mao et al, Sci Rep, 2015, 5, 16247). Wild-type and adult-treated $Asl^{Neo/Neo}$ mice, although heavier than untreated $Asl^{Neo/Neo}$ mice (Figure 3) are still performing significantly better than untreated $Asl^{Neo/Neo}$ mice. Taking into consideration the difference in body weight observed between tested groups, this does not affect the validity of our results. Indeed, the opposite relation between better rotarod performance and heavier body weight will have been a supplementary difficulty to prove the increased performance in wild-type and adult-treated $Asl^{Neo/Neo}$ mice. Untreated and neonatally-treated $Asl^{Neo/Neo}$ mice had similar weight (Figure 3). This has been added in the Results section.

With regards to open field test, the authors haven't found any report mentioning that this test could be affected by the body weight of tested mice.

9. Please indicate the age when cerebral NO metabolism is assessed in gene therapy-treated mice (Figure 5)

We thank the reviewer for this comment.

Age of mice at assessment was 9 or 13 months for WT, 1-13 months for untreated $Asl^{Neo/Neo}$ mice (as less than 10% of the animals had survived at the end of the study), 13 months for adult-injected $Asl^{Neo/Neo}$ mice, 9 months for neonatally-injected $Asl^{Neo/Neo}$ mice. This has been added for clarity in the legend of Figures 4 and 5.

10. In Figure 6, the authors evaluate apoptotic cells in the cortex. However, the age range of the untreated mice is 21 days to 12 months whereas the other groups are 9-12 months of age. It is unclear whether one can compare the brain of a 21 day old mouse with a mouse of 9-12 months of age. It is unclear why such a large age range in untreated mice was used.

We thank the reviewer for this comment. As presented in Supplementary Figure 7a, 75% of untreated $Asl^{Neo/Neo}$ mice not receiving a supportive treatment, which is described in Supplementary Figure 6a, have died by 3 weeks of age. To increase the number of untreated $Asl^{Neo/Neo}$ mice assessed for the experiment described in Figure 6, a large age range (3-6 weeks, n=3; 4 to 6 months, n=3; 9 to 12 months, n=3) was used to obtain enough samples for increasing the power of analysis at the time of experiment.

11. The authors showed that nitrite/nitrate levels are normalized in neonatally injected mice. Again, this is not a sufficient marker. What about cGMP level? Protein nitrosylation?

We thank the reviewer for this comment. To evaluate the efficacy of gene therapy in neonatally-injected $Asl^{Neo/Neo}$ mice, nitrite/nitrate levels are presented in Figure 5b.

Another method assessing protein nitrosylation with nitrotyrosine staining of the cortex and relative quantification by computerised method is presented in Figure 5c and 5d, respectively. Both methods show significant reduction in neonatally-injected $Asl^{Neo/Neo}$ mice compared to untreated and adult-injected $Asl^{Neo/Neo}$ mice.

cGMP levels have been measured and are presented in Supplementary figure 12. Untreated $Asl^{Neo/Neo}$ mice showed a significant increase compared to WT with persisting high levels in adult-treated $Asl^{Neo/Neo}$ mice and normalisation of levels in 2 out of 3 neonatally-injected $Asl^{Neo/Neo}$ mice but with overall no significant difference with untreated $Asl^{Neo/Neo}$ mice. This has been added in the result section.

No brain samples of treated *Asl*^{Neo/Neo} mice remained for further analysis of these, or other biomarkers.

12. Reduced liver glutathione was shown to be decreased in untreated ASLNeo/Neo mice (Supplementary Figure 8B). Was this changed in gene therapy-treated mice?

We thank the reviewer for this comment. This experiment has been repeated with liver samples stored from adult-injected and neonatally-injected *Asl*^{Neo/Neo} mice. No significant increase in reduced glutathione levels was observed in liver samples from untreated and treated groups compared to WT mice. These results are presented in Supplementary Figure 11b. This has been added in the result section.

13. It is unclear why the authors did not use an “empty” virus control in these experiments.

We thank the reviewer for this comment. The authors did not use an “empty” virus in the control group and these mice did not receive any injection of vector.

The first reason is that AAV is a known non-pathogenic virus and AAV vectors are well tolerated with no toxicity described in mouse models at the doses used in this study. Various experiments in mice have shown the absence of secondary effect associated with the AAV8 capsid for liver or CNS transduction. Several studies have been reported in mouse models of urea cycle defects using AAV gene therapy without using an empty vehicle injection in the control group (Cunningham et al, Mol Ther. 2009 Aug;17(8):1340-6. doi: 10.1038/mt.2009.88; Kok et al, Mol Ther. 2013 Oct;21(10):1823-31. doi: 10.1038/mt.2013.139; Chandler et al, Gene Ther. 2013 Dec;20(12):1188-91. doi: 10.1038/gt.2013.53).

A second reason is that the presence of DNA genome affects particle stability in AAV (Horowitz et al, J Virol, 2013, 87(6): 2994-3002.2013) and other viruses (Saha et al, Viruses, 2014; 6(9): 3563; Ivanovska et al, Proceedings of the National Academy of Sciences, 2007; 104(23): 9603-9608) therefore empty vector would not make a suitable control.

Minor comments:

1. The graphic of the urea cycle in Supplementary Figure 1 is confusing. For instance, citrulline is not included in the “urea cycle” portion of the graphic although citrulline is an intermediate in the urea cycle.

We thank the reviewer for this comment. This has been added in the amended Supplementary Figure 1.

2. On page 16, the authors state that hypoargininemia likely causes the hair phenotype. However, if this is the case, they do not speculate as to why this hair phenotype is unique to argininosuccinic aciduria as compared to other urea cycle disorders, which are also associated with low arginine levels.

We thank the reviewer for this comment. This hair phenotype is not restricted to this mouse model of argininosuccinic aciduria. This phenotype is also observed in Ornithine transcarbamylase (OTC) and argininosuccinic synthase (ASS) deficient mice, which are arginine deficient too (Shimada et al J Dermatol Sci. 1994 Jul;7 Suppl:S27-32. PubMed PMID: 7999674; Kok et al Mol Ther. 2013 Oct;21(10):1823-31. doi: 10.1038/mt.2013.139. PubMed PMID: 23817206). However this phenotype is not observed in the arginase deficient mice associated with hyperargininaemia (Cantero, J Neuroscience; 2016, 36, 6680-6690). As ASL deficiency is the enzymatic block directly impeding arginine synthesis, it is likely that arginine deficiency is more severe in ASL rather than more “proximal” urea cycle defects such as OTC or ASS deficiencies, in which the enzymatic deficiency is more upstream from the production of arginine from the catabolism of argininosuccinic acid.

3. In Figure 6(c), the line and the ‘ns’ sign comparing the statistical difference between WT and neonatally -injected mutants are partly missing.

We thank the reviewer for this comment. This has been corrected.

4. In Supplementary Figure 2, please indicate the age of mice when the brain tissues were collected.

This has been added in the amended figure, now Supplementary figure 4.

5. Figure 3: “(d, e) Mean growth of neonatally injected ASLNeo/Neo mice compared to WT (n=31) and untreated ASLNeo/Neo mice (n=41) (D) during the first month (n=13) and (e) over 9 months (n=7).” Letter (D) needs to be in lower case.

This has been corrected.

6. The authors should fix the numerous typographical errors throughout the manuscript.

Various typos have been found and corrected.

Reviewer #3 (Remarks to the Author):

The manuscript by Baruteau et al., describes that nitrosative/oxidative stress is increased in ASL-deficient mice, and that AAV-mediated expression of ASL in mice reverses pathological impairments caused by ASL deficiency. The former (i.e., nitrosative stress in the brain) is a novel finding, and the results are mostly convincing. However, the report overall is not informative and will only have minimal contribution to the field due to the reasons listed below. Thus, I feel that the report is not appropriate for Nature Communications.

1. Most experiments reported in this report show that AAV-mediated restoration of ASL expression in ASL deficient mice can correct the pathological phenotypes seen in the mice. These are completely expected results (since the authors are simply putting back the same gene that were removed from the mouse) and thus do not bring any new scientific information.

We thank the reviewer for this comment. This manuscript provides novel and not previously published important knowledge by describing the neurological phenotype of this mouse model and its pathophysiology. The interest of administering gene therapy at 2 different ages was to correct the 2 main affected organs responsible for the neurological phenotype: one approach when adult mutants were injected allows to target the liver, where the urea cycle is based. A second approach was to target the liver and the brain” (Neonatally-injected mutants). This allowed identification of a new pathophysiological mechanism of the brain disease with oxidative/nitrosative stress. This work helps to clarify i) a cause of the clinical paradox as to why argininosuccinic aciduria has a high rate of neurodisability despite a low rate of hyperammonaemic episodes, ii) a better design of novel therapeutic strategies to address the neurological complications of the disease.

In our opinion this manuscript adds valuable information for the scientific community as supported by other reviewers’ feedback.

2. ASL deficiency leads to decreased NO generation in liver, since ASL is coupled to NO synthase. In contrast, the authors claim that NO production and the cGMP signaling are upregulated in the brains of ASL-deficient mice. However, why ASL deficiency leads increased NO synthesis in the brain is not explained in the manuscript. The authors speculate that limited availability of GSH in neurons can cause increased nitrotyrosine staining, but the

limited GSH in the brain cannot explain why NO production (and cGMP formation) is increased in the brain.

We thank the reviewer for this interesting comment.

Oxidative stress has previously been associated with NOS uncoupling in *Asl^{Neo/Neo}* mice (Nagamani, 90(5):836-46. doi: 10.1016/j.ajhg.2012.03.018). Oxidative stress generates reactive oxygen species like peroxynitrite. Supported by previous publications (Gegg et al. J Neurochem. 2003 Jul;86(1):228-37. PubMed PMID: 12807442), we propose that the cerebral increase of NO levels is explained by the detoxification of peroxynitrite by reduced glutathione (GSH), which generate nitrite *via* the reaction $\text{ONOO}^- + 2\text{GSH} \rightarrow \text{NO}_2^- + \text{GSSG} + \text{H}_2\text{O}$ (This is discussed in the first paragraph of the Results section). Previous publications described increased cerebral levels of NO when exposed to oxidative stress (assessed similarly to our work by the Griess reaction) and decreased cerebral levels of reduced glutathione in mice Esmekaya et al. J Chem Neuroanat. 2016 Sep;75(Pt B):111-5. doi: 10.1016/j.jchemneu.2016.01.011. PubMed PMID: 26836107) or rats (Abd El-Gawad et al. J Biochem Mol Toxicol. 2004;18(2):69-77. PubMed PMID: 15122648).

Our work highlights the role of nitrosative stress (Nitrite/nitrate levels and nitrotyrosine staining) associated with oxidative stress and shows increased biomarkers (Nitrite/nitrate and cGMP levels). This concomitant increase of NO and cGMP levels suggests a persistence of the physiological NO/cGMP pathway with NO produced by coupled nitric oxide synthase (NOS).

The 2 mechanisms of NOS coupling and uncoupling are not exclusive and have been shown co-existing in various neurodegenerative disorders, as discussed in the first paragraph of the discussion.

These pathways are summarised in Figure 7.

3. It was very difficult for this reviewer to understand the experiment scheme for Figure 3 (animal injections and analysis) described in page 8. For instance, only 25% of untreated ASL KO mice survived after the injection; did the authors monitor the growth of survived animals for 12 month? Extensive editing is recommended for clarity.

We thank the reviewer for this comment. Yes, the growth and biomarkers levels were analysed in surviving untreated *Asl^{Neo/Neo}* mice. However these mice died progressively with only 25% of mice surviving after 1 to 2 months of age and 10% after 7-8 months of age.

For clarification, a paragraph entitled “Experimental design” has been added in the “Methods” section. A supplementary figure illustrating the experimental design has been added as Supplementary Figure 6.

4. In Figure 1, the authors need to conduct a co-staining experiment with a neuronal marker to prove that nitrotyrosine, nNOS, and iNOS are indeed increased in neurons.

We thank the reviewer for this comment. These complementary experiments have been performed and are presented in amended Figure 1 for nitrotyrosine and Supplementary Figure 3 for nNOS and iNOS.

5. Did “untreated” control Asl KO mice receive “vehicle” injections? Controls should be vehicle-injected animals.

We thank the reviewer for this comment. This question was answered earlier (Reviewer 2, Major comments, question 13).

The authors did not use an “empty” virus in the control group and these mice did not receive any injection of vector.

AAV is a known non-pathogenic virus and AAV vectors are well tolerated with no toxicity described in mouse models at the doses used in this study. Various experiments in mice have shown the absence of secondary effect associated with the AAV8 capsid for liver or CNS transduction. Several studies have been reported in mouse models of urea cycle defects using AAV gene therapy without using an empty vehicle injection in the control group (Cunningham et al, *Mol Ther.* 2009 Aug;17(8):1340-6. doi: 10.1038/mt.2009.88; Kok et al, *Mol Ther.* 2013 Oct;21(10):1823-31. doi: 10.1038/mt.2013.139; Chandler et al, *Gene Ther.* 2013 Dec;20(12):1188-91. doi: 10.1038/gt.2013.53).

A second reason is that the presence of DNA genome affects particle stability in AAV (Horowitz et al, *J Virol*, 2013, 87(6): 2994-3002.2013) and other viruses (Saha et al, *Viruses*, 2014; 6(9): 3563; Ivanovska et al, *Proceedings of the National Academy of Sciences*, 2007; 104(23): 9603-9608) therefore empty vector would not make a suitable control.

Minor concerns

i) Figure citation is not in a Nature Communications format (for instance, Fig. 1A should be Fig 1a).

We thank the reviewer for this comment. This has been amended as per Nature Communications format.

ii) In Fig. 2, EFS is not explained in the figure legend. I think it should be labeled “AAV” or “GFP” in this Figure.

We thank the reviewer for this comment. A GFP label has been inserted in the amended Figure 2 and Supplementary Figure 5.

iii) Figure 3a and 3b are not cited in the text.

We thank the reviewer for this comment. This has been corrected.

scAAV2/9 iv Neonatal scAAV2/8 iv Neonatal Uninjected

Annex 1. Neonatal intravenous injection of titre-matched AAV vectors encoding the GFP reporter gene in CD-1 pups.

Reviewers' comments:

Reviewer #1 (Remarks to the Author):

Baruteau et al present a revised manuscript. The observations that fixing the brain NO axis is critical for ASL deficiency is further supported by the data here, and proven by experimentation. Overall the results are important and the paper is well written, but has limited new data, but no new experiments with neuronotrophic AAVs configured with CNS specific promoters as suggested. A practical rationalization of why not to do the studies is presented which is debatable at best. The authors provide commentary on why this is could be difficult, and the reviewer understands that additional gene therapy studies in mice are not trivial. However, the concept of leakage is misleading as others have surmounted similar challenges using clean CNS promoters in AAVs. Serotypes can help accomplish this – ei PHP.B. A vector using a CNS promoter with a PHP.B serotype (or AAV9 or even AAV8) could easily be constructed and given to mice to study effects of CNS correction +/- acylation. What dominates the pathophysiology and symptoms could be addressed using such a model. The PHP.B could help with the murine studies and whether it would make it to the clinic is uncertain, but it would provide precedence for the use of perhaps a similar performing serotype. Commentary is also provided on the CNS tropism of AAV8 but to the knowledge of this reviewer, AAV8 for CNS gene therapy, especially as a systemic injection, has not been performed in humans. The serotypes of interest that have been given to humans include AAV9 (SMA) and rh10 (Batten disease). AAV2 and perhaps AAV5 might be on this list as well, but certainly these serotypes are not as potent as AAV9 and rh10 for CNS gene therapy. The point here is that while AAV8 is promising in mice (most serotypes are) and perhaps the fetal primate for CNS delivery, it is not close to the clinic for CNS delivery (hepatic delivery of AAV8 is established) as the authors suggest. If I am mistaken, please offer the references of papers or trials that treat humans with AAV8 for a CNS indication. If I am in error, this latter point of lack of translation is not relevant.

The authors may be aware of a recent paper describing the existence of a cryptic hepatic enhancer very close to the ITRs (Logan GJ, ... Alexander IE. Identification of liver-specific enhancer-promoter activity in the 3' untranslated region of the wild-type AAV2 genome. *Nat Genet.* 2017 Aug;49(8):1267-1273. doi: 10.1038/ng.3893. Epub 2017 Jun 19. PubMed PMID: 28628105). In fact, this element was discovered because it confers hepatic expression on an empty cassette (see figures in paper). As Logan et al note, these elements have been propagated in many AAV vectors and this reviewer wonders whether the authors' hSyn AAVs carry this element or part of it? It could easily explain hepatic expression from a CNS promoter in the context of an AAV vector. Given what is now known, and published with some CNS promoters showing strict expression, it should be routine to swap the Asl cDNA into another vector, make new AAVs, especially an AAV8 to support the claims of this paper, and show it is exclusively expressed in the CNS with? effects on phenotype. The current vector uses a traditional EFs promoter and WPRE to direct wide expression in the mouse after high dose systemic delivery of an AAV8 vector. Whether it could be translated, even if configured with a human cDNA, is questionable. My suggestion remains to the authors to make an AAV for CNS gene therapy only – it can be done and made to not leak, and test it in the Asl mice to ascertain the importance of CNS correction (NO pathway) in this disorder.

Reviewer #2 (Remarks to the Author):

1) I'm not sure how Supplementary Figure 3 b, d, and f show that there is a significant increase of nNOS, iNOS, or eNOS and by immunofluorescence in neurons/ brain endothelial cells. These figures simply show that nNOS and iNOS were expressed in neurons and eNOS was expressed in the endothelial cells. To show that these NOS isoforms are increased, double staining of NOS and cell marker needs to be performed on both WT and Asl neo/neo .

2) "cGMP levels in brain homogenates was higher in adult-treated mice and normalised in 2 out of 3 samples tested although this did not reach significance (Supplementary Fig. 12)". This sentence is confusing and does not seem to correlate with the data. What are the authors specifically comparing? Also, the data showed that it is the neonatally-injected mice which were normalized in 2 out of 3 samples tested.

3) The lack of a vehicle only control group is a significant experimental design flaw.

4) Supplemental Figure 9 has a typographical error. C. has citrulline in figure but arginine in figure legend.

5) They still have several grammatical errors throughout that should be fixed.

Reviewer #3 (Remarks to the Author):

The authors have greatly improved the manuscript. A few minor concerns still remain.

1. For a broad range of readers, I think it would be helpful if the authors can further emphasize and explain the importance of their findings in the manuscript.

2. The author response regarding the increased levels of NO and cGMP in the ASL deficient mice is not sufficient. The authors responded that i) nitrite generation from peroxynitrite by GSH, and ii) NO production by coupled NOS could explain the increase in NO and cGMP levels. However, GSH only contributes to nitrite formation (not NO). In addition, ASL deficiency causes NOS uncoupling. I understand that both coupled and uncoupled NOS are present under neurodegenerative conditions, but why is the coupled NOS pathway activated in the ASL deficient mice? Can the authors please explain further about this in the manuscript?

We would like to thank you for your 2nd review about our manuscript NCOMMS-17-12400 entitled “Argininosuccinic aciduria fosters neuronal nitrosative stress reversed by Asl gene transfer”.

We are grateful to the reviewers for their comments as we are convinced that the first review significantly improved the quality of the manuscript. We have now carefully assessed the reviewers’ comments of this 2nd review, which we have addressed below. We would like to get clarification of the in vivo experiments requested by reviewer 1.

Please find your comments with our answers **in red** below and highlighted **in red** in the revised manuscript.

Reviewers' comments:

Reviewer #1 (Remarks to the Author):

Baruteau et al present a revised manuscript. The observations that fixing the brain NO axis is critical for ASL deficiency is further supported by the data here, and proven by experimentation. Overall the results are important and the paper is well written, but has limited new data, but no new experiments with neuronotrophic AAVs configured with CNS specific promoters as suggested. A practical rationalization of why not to do the studies is presented which is debatable at best. The authors provide commentary on why this is could be difficult, and the reviewer understands that additional gene therapy studies in mice are not trivial. However, the concept of leakage is misleading as others have surmounted similar challenges using clean CNS promoters in AAVs. Serotypes can help accomplish this – ei PHP.B. A vector using a CNS promoter with a PHP.B serotype (or AAV9 or even AAV8) could easily be constructed and given to mice to study effects of CNS correction +/- acylation.

We disagree with Reviewer #1 regarding the use of PHP.B as a specific target of the mouse CNS. For example Hordeaux and colleagues (Hordeaux et al, 2018; 26(3):664-8) unequivocally demonstrate expression of GFP following AAV.PHP.B to adult C57Bl/6 and Balb/C mice (see below).

What dominates the pathophysiology and symptoms could be addressed using such a model. The PHP.B could help with the murine studies and whether it would make it to the clinic is

uncertain, but it would provide precedence for the use of perhaps a similar performing serotype. Commentary is also provided on the CNS tropism of AAV8 but to the knowledge of this reviewer, AAV8 for CNS gene therapy, especially as a systemic injection, has not been performed in humans. The serotypes of interest that have been given to humans include AAV9 (SMA) and rh10 (Batten disease). AAV2 and perhaps AAV5 might be on this list as well, but certainly these serotypes are not as potent as AAV9 and rh10 for CNS gene therapy. The point here is that while AAV8 is promising in mice (most serotypes are) and perhaps the fetal primate for CNS delivery, it is not close to the clinic for CNS delivery (hepatic delivery of AAV8 is established) as the authors suggest. If I am mistaken, please offer the references of papers or trials that treat humans with AAV8 for a CNS indication. If I am in error, this latter point of lack of translation is not relevant.

We have emphasised that AAV8 is likely not the serotype of choice for clinical translation, particularly as clinical trials for spinal muscular atrophy using AAV9 show substantial efficacy. We have added a sentence in the last paragraph of the discussion highlighting the need of optimisation of the vector before clinical translation.

The authors may be aware of a recent paper describing the existence of a cryptic hepatic enhancer very close to the ITRs (Logan GJ, Alexander IE. Identification of liver-specific enhancer-promoter activity in the 3' untranslated region of the wild-type AAV2 genome. *Nat Genet.* 2017 Aug;49(8):1267-1273. doi: 10.1038/ng.3893. Epub 2017 Jun 19. PubMed PMID: 28628105). In fact, this element was discovered because it confers hepatic expression on an empty cassette (see figures in paper). As Logan et al note, these elements have been propagated in many AAV vectors and this reviewer wonders whether the authors' hSyn AAVs carry this element or part of it? It could easily explain hepatic expression from a CNS promoter in the context of an AAV vector. Given what is now known, and published with some CNS promoters showing strict expression, it should be routine to swap the Asl cDNA into another vector, make new AAVs, especially an AAV8 to support the claims of this paper, and show it is exclusively expressed in the CNS with? effects on phenotype.

We are very familiar with the publication. Although a liver enhancer may account for some liver expression following delivery of an AAV carrying the Synapsin promoter, it is unlikely to account for the expression which we observe, following administration of such a vector, in other non-neuronal tissues (unpublished data, below).

The current vector uses a traditional EFs promoter and WPRE to direct wide expression in the mouse after high dose systemic delivery of an AAV8 vector. Whether it could be translated, even if configured with a human cDNA, is questionable.

EFS and WPRE have been successfully used in a clinical trial for SCID-X1 with a modified γ -retroviral vector (Hacein-Bey-Abina et al, N Engl J Med, 2014; 371(15):1407-17. We agree with the reviewer in that there is no clinical precedent for liver and CNS applications.

My suggestion remains to the authors to make an AAV for CNS gene therapy only – it can be done and made to not leak, and test it in the Asl mice to ascertain the importance of CNS correction (NO pathway) in this disorder.

We thank the reviewer for his comment. We would first like to restate that, from our own and published data, we cannot guarantee the avoidance of non-neuronal expression, either by choice of capsid configuration or promoter. Therefore, we contend that such experiments would be meaningless. Secondly, if at some point such a vector could be constructed, it's use in the experiment suggested by the reviewer would be very unlikely to provide further insight into disease pathogenesis. Namely, ASL deficient mice receiving such a vector would theoretically express ASL in brain, but not liver, and therefore the neurotoxicity caused by hyperammonemia would obscure any local neuronal correction. We maintain that the experiments performed in this study are the best currently possible to interrogate the underlying pathology of this model, and secondly serve as proof of concept for clinical translation with a vector more suitable for human translation.

Reviewer #2 (Remarks to the Author):

1) I'm not sure how Supplementary Figure 3 b, d, and f show that there is a significant increase of nNOS, iNOS, or eNOS and by immunofluorescence in neurons/ brain endothelial cells. These figures simply show that nNOS and iNOS were expressed in neurons and eNOS was expressed in the endothelial cells. To show that these NOS isoforms are increased, double staining of NOS and cell marker needs to be performed on both WT and *As1 neo/neo* .

We thank the reviewer for his comment. Increased 3',3 diaminobenzidine (DAB) staining of nNOS, iNOS and eNOS in *As1^{Neo/Neo}* mice versus wild-type is presented in supplementary figures 3a, 3c and 3e, respectively. Morphological analysis was suggesting a cellular localisation in neurons for nNOS and iNOS and in endothelial cells for eNOS, which is confirmed by immunofluorescence in *As1^{Neo/Neo}* mice in supplementary figures 3b, 3d and 3f respectively.

2) "cGMP levels in brain homogenates was higher in adult-treated mice and normalised in 2 out of 3 samples tested although this did not reach significance (Supplementary Fig. 12)". This sentence is confusing and does not seem to correlate with the data. What are the authors specifically comparing? Also, the data showed that it is the neonatally-injected mice which were normalized in 2 out of 3 samples tested.

Thank you for this comment. The sentence in the text has been amended for better clarity.

3) The lack of a vehicle only control group is a significant experimental design flaw.

We thank the reviewer for this comment.

As discussed in our first rebuttal letter, we did not use a "vehicle" vector in the control group for the following reasons:

- AAV is a non-pathogenic virus. Various experiments in mice have shown the absence of secondary effect associated with the AAV8 capsid for liver or CNS transduction at the doses used in this study. Several studies have been reported in mouse models of urea cycle defects using AAV gene therapy without using an empty vehicle injection in the control group (Cunningham et al, Mol Ther. 2009 Aug;17(8):1340-6. doi: 10.1038/mt.2009.88; Kok et al,

Mol Ther. 2013 Oct;21(10):1823-31. doi: 10.1038/mt.2013.139; Chandler et al, Gene Ther. 2013 Dec;20(12):1188-91. doi: 10.1038/gt.2013.53).

- The presence of DNA genome affects particle stability in AAV (Horowitz et al, J Virol, 2013, 87(6): 2994-3002.2013) and other viruses (Saha et al, Viruses, 2014; 6(9): 3563; Ivanovska et al, Proceedings of the National Academy of Sciences, 2007; 104(23): 9603-9608) therefore empty vector would not make a suitable control.

4) Supplemental Figure 9 has a typographical error. C. has citrulline in figure but arginine in figure legend.

Thank you. This has been corrected.

5) They still have several grammatical errors throughout that should be fixed.

Thank you. The manuscript has been read again extensively by various native-English speakers who have refined formulation of sentences.

If the reviewer can still find some grammatical errors, can he let us know what errors he is still referring to?

Reviewer #3 (Remarks to the Author):

The authors have greatly improved the manuscript. A few minor concerns still remain.

1. For a broad range of readers, I think it would be helpful if the authors can further emphasize and explain the importance of their findings in the manuscript.

We thank the reviewer for this comment.

Additional sentences have been added in the introduction and the discussion to highlight the findings of this study and their clinical relevance, which helps to better understand the neurological phenotype of argininosuccinate lyase deficiency and the inefficient therapeutic strategy of a liver-restricted therapy to preserve the neurocognitive function of patients affected by ASL deficiency.

2. The author response regarding the increased levels of NO and cGMP in the *Asl* deficient mice is not sufficient. The authors responded that i) nitrite generation from peroxynitrite by GSH, and ii) NO production by coupled NOS could explain the increase in NO and cGMP levels. However, GSH only contributes to nitrite formation (not NO). In addition, ASL deficiency causes NOS uncoupling. I understand that both coupled and uncoupled NOS are present under neurodegenerative conditions, but why is the coupled NOS pathway activated in the ASL deficient mice? Can the authors please explain further about this in the manuscript?

We thank the reviewer for this comment.

NO has a very short half-life of few seconds and is reliably assessed by downstream metabolites (nitrite and nitrate), which are more stable (as mentioned in the first paragraph of the Result section and supported by reference 20).

Asl^{Neo/Neo} mice are hyperammonaemic. Physiological response to hyperammonaemia in the brain causes an increase of cerebral glutamate levels, which activates NMDA receptors and triggers the physiological NO/cGMP pathway via activation of nNOS and iNOS (Reference 26). This glutamate/NO/cGMP pathway, which requires coupled NOS, is observed in vitro (Rodrigo et al, *Neurobiol Dis*, 2005; 19(1-2):150-61. PMID: 15837570) and in vivo (Rodrigo et al, *J Neurochem*, 2007; 102(1):51-64. PMID: 17286583).

NOS uncoupling is favoured by low arginine in tissue but both NOS coupling and uncoupling co-exist in tissues of *Asl^{Neo/Neo}* mice as shown by Nagamani et al (Reference 22). NOS monomers are not all uncoupled in low arginine conditions and therefore a persisting physiological NO production is observed whilst NOS uncoupling generates reactive oxygen species in *Asl^{Neo/Neo}* mice (References 16 and 22).

This part of the discussion has been made clearer and this proposed pathophysiology of the brain disease is summarised in Figure 7.

Reviewers' comments:

Reviewer #1 (Remarks to the Author):

Baruteau et al present a second revised manuscript. There has been some banter between the reviewer and authors surrounding the need for CNS correction and translational of the results. The best set of studies in these mice, which are much, much more robust than some other UCD mutants (except Spf Ash), would be to use hepatic vs cerebro-hepatic vs cerebro correction and compare the results. My initial suggestion was to use the same serotype configured with a neuronal promoter to compare such effects, especially since it has been well recognized by others over the years that the NO axis is perturbed in ASuria. The authors cite a recent paper on PHP.B but have totally missed the point of my commentary which I will try to re-clarify here. The suggestion is to construct a neuronotrophic AAV with a characterized CNS specific promoter to direct robust and widespread ASL expression- PHPB was suggested because it appears more potent at CNS penetration via the capsid. Obviously such as cassette with an EF1 promoter will express everywhere. AAV8 would also be fine to use based on the results presented by the authors but I believe it will be much less potent than others. The experiments that still need to be performed would be to construct an AAV vector configured as follows: AAV2/(8 or PHPB or rh8 or rh10 or other NEUROTROPHIC CAPSID) ITR-CNS specific promoter_+/- intron ASL +/- WPRE + polyA - ITR and deliver this to the mutant mice vs the Ef1 cassette, asking the question how much can CNS (aka NO axis) correction improve the mice even without acylation/arg or liver therapy? The survival curves clearly suggest about 20% of the mice live to the age of 1 year with no therapy. Some patients with ASL have minimal/mild hyperammonaemia so the relevance here is practical. Would neonatal or adult CNS directed (via systemic delivery) AND restricted expression (via the promoter +/- miRNA NOT capsid) prolong survival or improve the phenotype seen in the uncorrected mice? Would the NO axis rescue dominate mild hyperammonaemia? Acylation therapy could be added to show that the mice could have improved neurocognitive function when CNS only expression of ASL was restored vs non AAV treated mice.

To reiterate: the reason PHPB was suggested is that the degree of CNS correction after AAV8 therapy in mice is not as great as AAV9, or PHPB, or perhaps rh8 or rh10. The CNS restriction of transgene expression would be through the promoter and miRNA as needed NOT the capsid. There are numerous small CNS promoters that have been studied (MECP2, Syn, CamKII, etc), as well as liver miRNA detargeting sequences that could be incorporated into the vector. There are even engineered serotypes that preferentially target the CNS over the liver after systemic delivery (Samulski lab has generated these capsids). Similarly, there are numerous liver specific promoters that could use to perform the analogous study - does isolated liver correction rescue the CNS NO deficit? This study might be more relevant because if isolated liver correction is as effective as hepato-CNS correction, the path to the clinic is clear for hepatic therapy - an additional CNS effect might not be needed which would lower doses and allow the use of tested liver cassettes.

The authors have also not answered the query about whether the cassette has the AAV HCC elements in it, yet do concede that a similarly configured AAV has never been given to humans - specifically an AAV vector with the WPRE element. They provide an example of integrating vectors in an immunodeficiency syndrome but my query was clearly directed toward AAV gene therapy for hepatic correction with AAV vectors containing a WPRE. The fact that the serotype, cassette, and other elements are not translatable YET also seem to be accepted bilaterally.

Remaining concerns:

There are also concerns about mouse numbers for critical studies (cGMP measurement line 268-270, survival line 1042 n=5 for adult treated mice), in places where enzyme activity is measured there are no ranges (see lines 255 ASL activity in mutant 14.1 % vs control 16.2 % - need ranges and stats here - enzyme assays are usually performed in triplicate and preferable repeated three times, here it is duplicate but ranges are not reported) and the claims about this approach for hepatocerebral correction after systemic AAV therapy are not novel. This has been claimed in LSDs after AAV.

I stated this before ; please correct this sentence(line 172 we designed a gene therapy approach to normalise ammoniaemia and

conditionally target neuronal ASL activity). THIS STUDY HAS NOT DEMONSTRATED CONDITIONAL NEURONAL TARGETING.

I enjoyed reading the paper and all the extensive details presented but for most readers, I think moving all the methods to supplementals with associated references, and cutting the discussion could reduce this from larger paper to a letter without impacting the message.

If the paper had the other AAV arms or perhaps cre studies as this is a hypomorphic allele (ie liver specific correction on synCRE ASL mouse ... others), not a null, there would be enough for a larger paper.

Despite the above caveats, and as stated previously, this is an important set of gene therapy studies for ASauria, which in humans is still essentially untreatable unless an elective LT pre cirrhosis is pursued, and such a transplant is experimental surgery. Newer insights are also provided in line with predictions about the NO axis in ASauria and agree with previous work from Dr Lees lab at Baylor in mice and pts.

Reviewer #2 (Remarks to the Author):

The authors have responded to this reviewer's comments. While I appreciate the references cited regarding the appropriateness of empty vector control in other models, the use of untreated mice as controls is inappropriate from a design and biological perspectives. There are now multiple examples where vehicle (in this case AAV infection) can have biological effects in translational studies. Still, in spite of this poor biological design, other conclusions of this manuscript is supported by the data. There are still significant English/typographical errors to be corrected.

Reviewer #3 (Remarks to the Author):

The manuscript has greatly improved. I have no further comments to add.

Reviewers' comments:

Reviewer #1

Baruteau et al present a second revised manuscript. There has been some banter between the reviewer and authors surrounding the need for CNS correction and translational of the results.

The best set of studies in these mice, which are much, much more robust than some other UCD mutants (except Spf Ash), would be to use hepatic vs cerebro-hepatic vs cerebro correction and compare the results.

My initial suggestion was to use the same serotype configured with a neuronal promoter to compare such effects, especially since it has been well recognized by others over the years that the NO axis is perturbed in ASauria.

The authors cite a recent paper on PHP.B but have totally missed the point of my commentary which I will try to re-clarify here.

The suggestion is to construct a neuronotrophic AAV with a characterized **CNS specific promoter** to direct robust and widespread ASL expression– PHPB was suggested because it appears more potent at CNS penetration via the capsid. Obviously such as cassette with an EF1 promoter will express everywhere. AAV8 would also be fine to use based on the results presented by the authors but I believe it will be much less potent than others. The experiments that still need to be performed would be to construct an AAV vector configured as follows: AAV2/(8 or PHPB or rh8 or rh10 or other NEUROTROPHIC CAPSID) ITR-**CNS specific promoter**_+/- intron ASL +/- WPRE + polyA – ITR and deliver this to the mutant mice vs the Ef1 cassette, asking the question how much can CNS (aka NO axis) correction improve the mice even without acylation/arg or liver therapy? The survival curves clearly suggest about 20% of the mice live to the age of 1 year with no therapy.

Some patients with ASL have minimal/mild hyperammonaemia so the relevance here is practical.

Would neonatal or adult CNS directed (via systemic delivery) AND restricted expression (via **the promoter** +/- miRNA NOT capsid) prolong survival or improve the phenotype seen in the uncorrected mice? Would the NO axis rescue dominate mild hyperammonaemia? Acylation therapy could be added to show that the mice could have improved neurocognitive function when CNS only expression of ASL was restored vs non AAV treated mice.

To reiterate: the reason PHPB was suggested is that the degree of CNS correction after AAV8 therapy in mice is not as great as AAV9, or PHPB, or perhaps rh8 or rh10. The CNS restriction of transgene expression would be through **the promoter** and miRNA as needed NOT the capsid. There are numerous small CNS promoters that have been studied (MECP2, Syn, CamKII, etc), as well as liver miRNA detargeting sequences that could be incorporated into the vector. There are even engineered serotypes that preferentially target the CNS over the liver after systemic delivery (Samulski lab has generated these capsids).

Similarly, there are numerous liver specific promoters that could use to perform the analogous study – does isolated liver correction rescue the CNS NO deficit?

This study might be more relevant because if isolated liver correction is as effective as hepato-CNS correction, the path to the clinic is clear for hepatic therapy – an additional CNS effect might not be needed which would lower doses and allow the use of tested liver cassettes. The authors have also not answered the query about whether the cassette has the AAV HCC elements in it, yet do concede that a similarly configured AAV has never been given to humans – specifically an AAV vector with the WPRE element. They provide an example of integrating vectors in an immunodeficiency syndrome but my query was clearly directed toward AAV gene therapy for hepatic correction with AAV vectors containing a WPRE. The fact that the serotype, cassette, and other elements are not translatable YET also seem to be accepted bilaterally.

We thank the reviewer for their extensive discussion. Our current work present cohort with ASL hepatic correction (adult-injected mutants) versus ASL cerebro-hepatic correction (neonatally-injected mutants). As stated in our extensive responses in the past two rounds of review, there are technical and logistical reasons why the reviewer's suggestion (experiment with ASL cerebral correction only) is untenable.

- (1) This experiment is irrelevant to patients and **has no clinical interest**. Targeting the cerebral disease only and not the urea cycle is unrealistic. 90% of patients require conventional treatment (protein-restricted diet and ammonia scavengers) (reference 19). If this standard of care is stopped, patients will experience hyperammonaemic decompensations.
- (2) To perform the experiment described by reviewer 1 and characterise the neurological disease when the NO axis only is corrected, requires a vector transducing the CNS only with no liver expression. In the hypomorphic ASL-deficient mouse model, mutants are hyperammonaemic despite a high liver ASL residual activity (14.5% of WT). Neonatally-injected mutants have normal plasma ammonia levels with a 18.5% liver ASL activity of WT (See Results and Figure 4). Therefore even a small percentage of liver transduction might reduce plasma ammonia levels and generate bias on the respective benefit of the urea cycle effect or the rescue of NO axis. Novel engineered capsids (like Wang et al, Mol Ther Methods Clin Dev, 2018;9:234-246) and miRNA de-targeting (Xie et al, Mol Ther, 2011;19(3):526-535) reduce liver expression but can not guarantee **no** hepatocyte transduction.

In our experience, a truncated promoter, in the context of an AAV backbone, cannot be guaranteed to achieve tight restriction of expression to neurons For example, we performed

an experiment testing a synapsin promoter (neuronal promoter) in a neurotropic AAV9 administered intravenously. We observed marker gene expression not only in brain, but also liver, heart and kidney (below). Knowing this, and in order to meet the reviewers request, we would need to perform extensive marker gene characterisation of a putative neuron-specific promoter before even embarking on a therapeutic approach and without guarantee of success.

(3) Another issue is to ensure a normalisation of ammonia levels prior to withdrawing diet and ammonia scavengers to avoid chronic exposure to mild hyperammonaemia, which will cause hyperammonaemia-related neurotoxicity. Ammonia is extremely neurotoxic and increases cerebral glutamate and nitric oxide levels and triggers oxidative stress (cf discussion, reference 26 and Fig. 7). pharmaceutical and dietary support provided to these *Ast^{Neo/Neo}* mice (based upon previously-published maintenance regimes) lowers, but does not normalise blood ammonia levels. Despite this supportive treatment 25% of mutant mice die before day 30 (Supplementary Fig.7a). Neither does this treatment improve the weight gain of these mice (Supplementary Fig. 7b). Therefore, if the experiment was performed as per the reviewer's suggestion, omitting central metabolic correction would cause unchecked glutaminergic neurotoxicity, which will bias results and prevent to draw any relevant conclusion. Failure to normalise hyperammonaemia and its modifications of the NO pathway would have an overwhelming confounding influence.

The experiments would require large groups of mice, since half of uncorrected and CNS-treated animals would die within 5 days from hyperammonaemia (Fig. 3a). Not only would these experiments be technically difficult (in terms of time taken to generate such groups), but such experiments would be inconsistent with the principles of the 3Rs in that they would be

uninformative with excess wastage of animal lives, and would cause unnecessary suffering once urea cycle support was withdrawn.

In his comment, reviewer 1 raises the question “*does isolated liver correction rescue the CNS NO deficit?*”. The authors are worried that their manuscript has not been well understood by the reviewer as this is the key take-home message of our manuscript. Our current work presents hepatic (adult-injected mutants) versus cerebro-hepatic (neonatally-injected mutants) rescue of the disease. This clearly demonstrates that isolated liver correction does not rescue the CNS NO deficit (Figures 5 and 6). Hepatic therapy alone (as currently does liver transplantation) rescues the urea cycle but does not correct the neurological disease.

We have emphasised in our previous rebuttal letters several times that if this work provides a proof-of-concept that AAV vector can rescue the liver and brain disease in ASA, this vector by no means will be translated to humans. We all agree that this vector needs more work before clinical translation. There is a sentence mentioning it in the last paragraph of the discussion. EFS promoter and WPRE sequence have been used in clinical trial with a different (γ -retroviral) vector and targeting haematopoietic stem cells (Hacein-Bey-Abina et al, N Engl J Med, 2014; 371(15):1407-17), i.e. highly proliferative cells with integrating vector, which therefore claims even better for a safety of these sequences in humans. We agree with the reviewer in that there is no clinical precedent for liver and CNS applications.

Regarding the remaining concerns for reviewer 1, please find our responses below.

Remaining concerns:

There are also concerns about mouse numbers for critical studies (cGMP measurement line 268-270, survival line 1042 n=5 for adult treated mice),

This cGMP analysis was performed as requested at the first review on the remaining 4 brain samples from 5 adult-treated mutants.

in places where enzyme activity is measured there are no ranges (see lines 255 ASL activity in mutant 14.1 % vs control 16.2 % - need ranges and stats here –

We have now added ranges to the text. Statistics are provided in Figures 4 and 5 to compare all groups together regarding liver and brain ASL activity, respectively.

enzyme assays are usually performed in triplicate and preferable repeated three times, here it is duplicate but ranges are not reported)

Enzyme activities are sometimes performed in duplicate (Yang et al, Nature Biotech, 2016;34(3):334-340). We performed our analyses in duplicates as constrained by the availability of material and reagents. We have added ranges in Results.

and the claims about this approach for hepatocerebral correction after systemic AAV therapy are not novel. This has been claimed in LSDs after AAV.

We agree with the reviewer. We are not claiming it is novel. We are stating in our abstract that “this approach provides new hope for hepatocerebral metabolic diseases”, meaning that this study reinforces conclusions from previous publications proposing a systemic gene therapy approach for treating hepatocerebral disease in LSD.

I stated this before ; please correct this sentence(line 172 we designed a gene therapy approach to normalise ammoniaemia and conditionally target neuronal ASL activity). THIS STUDY HAS NOT DEMONSTRATED CONDITIONAL NEURONAL TARGETING.

We apologise for failing to address the reviewer’s comment previously – we agree and have removed the word “conditionally”.

I enjoyed reading the paper and all the extensive details presented but for most readers, I think moving all the methods to supplementals with associated references, and cutting the discussion could reduce this from larger paper to a letter without impacting the message. If the paper had the other AAV arms or perhaps cre studies as this is a hypomorphic allele (ie liver specific correction on synCRE ASL mouse ... others), not a null, there would be enough for a larger paper.

We thank the reviewer for these positive comments here.

The manuscript presents a large amount of data and has already 12 supplemental figures. Various supplementary experiments and commentaries have been requested by this and other reviewers previously, which have contributed overall to a more detailed and longer paper. However all these addendums have been made in the interest of clarity and precisions for the reader and have satisfied the 2 other reviewers.

We are surprised that reviewer 1, who had very positive comments initially in his first review (July 2017), who has acknowledged significant improvement of the manuscript in his 2nd review (February 2018), is now proposing in this 3rd review to shorten the manuscript in a letter format. We will await guidance from the editorial team.

The suggestion of using a Cre mice to abolish Asl expression in neurons would provide an interesting experimental design to explore in more details the neurological disease of ASA. The authors have already considered this approach. However adding this experiment in the current manuscript will be by far beyond the objectives of the current set of experiments and will require another manuscript in itself to interpret the pathophysiological mechanisms.

Despite the above caveats, and as stated previously, this is an important set of gene therapy studies for ASAAuria, which in humans is still essentially untreatable unless an elective LT pre cirrhosis is pursued, and such a transplant is experimental surgery. Newer insights are also provided in line with predictions about the NO axis in ASAAuria and agree with previous work from Dr Lees lab at Baylor in mice and pts.

We thank the reviewer for acknowledging the importance of this manuscript in the field of ASAAuria, in which we interrogate a novel pathophysiological mechanism of the brain disease, independent of hyperammonaemia, and provide evidence of the necessity to target both the urea cycle (i.e. the liver) and the cerebral NO-citrulline cycle to correct the brain disease. This supports previous publications by the Baylor group.

Reviewer #2

The authors have responded to this reviewer's comments. While I appreciate the references cited regarding the appropriateness of empty vector control in other models, the use of untreated mice as controls is inappropriate from a design and biological perspectives. There are now multiple examples where vehicle (in this case AAV infection) can have biological effects in translational studies. Still, in spite of this poor biological design, other conclusions of this manuscript is supported by the data.

We are sorry that our experiment design is considered to be poor. We took the experimental design very seriously and considered whether the use of a control AAV would be a useful control. As detailed in our previous rebuttal letters, and acknowledged in our manuscript text, we decided that the following would be inappropriate for example i) *Empty AAV*. DNA genome affects particle

stability in AAV (Horowitz et al, J Virol, 2013, 87(6): 2994-3002.2013) and other viruses (Saha et al, Viruses, 2014; 6(9): 3563; Ivanovska et al, Proc Natl Acad Sci USA, 2007; 104(23): 9603-9608). Therefore, empty AAV is unlikely to traffic in the same way as particles containing genomes
ii) *AAV containing inactive ASL*. Since ASL forms a metabolic complex, presence of folded, but non-functional ASL might still associate with protein partners, and actually interfere with the residual ASL activity in these mice.

Numerous preclinical rodent experiments have been reported in mouse models of urea cycle defects using AAV gene therapy without using an empty vehicle injection in the control group (Cunningham et al, Mol Ther. 2009 Aug;17(8):1340-6. doi: 10.1038/mt.2009.88; Kok et al, Mol Ther. 2013 Oct;21(10):1823-31. doi: 10.1038/mt.2013.139; Chandler et al, Gene Ther. 2013 Dec;20(12):1188-91. doi: 10.1038/gt.2013.53). Translational studies are usually performed against controls injected with PBS versus empty vehicle (Greig et al, Hum Gene Ther Clin Dev, 2017;28(1):28-38; Lal et al, Hum Gene Ther Clin Dev, 2017; 27(4):145-151; Meadows et al, Hum Gen Ther Clin Dev, 2017;26(4):228-242.

There are still significant English/typographical errors to be corrected.

Thank you. The manuscript has been read extensively by various native-English speakers who have refined the formulation of sentences.

Reviewer #3

The manuscript has greatly improved. I have no further comments to add

We thank the reviewer for their time in the first two rounds of review

REVIEWERS' COMMENTS:

Reviewer #1 (Remarks to the Author):

Baruteau et al present continued commentary on the manuscript. Some points were well articulated, such as whether these studies demonstrate conditional genetics (NO!), while others continued to be debated, such as whether a clean expressing CNS AAV vector can be constructed (the answer here is yes, it can). Others have been outright ignored, despite multiple requests for commentary, such as whether the vectors contain the small genomic AAV hepatic enhancers that Dr Alexandra's group has identified. If so, they should probably mention this in the methods section. The comments here are particularly relevant for the synapsin promoter studies discussed in the last rebuttal. Despite this not being as a complete a story as expected from a group with such deep expertise in gene therapy, it has evolved and my initial impression that this paper represents an important practical demonstration of AAV gene therapy for Asl deficiency, a condition that features hyperammonaemia as well as NO axis CNS perturbations (that have been well described here for the first time), still stands. The manuscript has been significantly improved and aside from very minor corrections, such as the language surrounding conditional correction, is suitable to move forward in the publication process.

Reviewer #2 (Remarks to the Author):

The authors have addressed the concerns of this reviewer. While the manuscript and data could have been improved from a design perspective, I think the data support the general conclusions.

Reviewers' comments:

Reviewer 1

Baruteau et al present continued commentary on the manuscript. Some points were well articulated, such as whether these studies demonstrate conditional genetics (NO!), while others continued to be debated, such as whether a clean expressing CNS AAV vector can be constructed (the answer here is yes, it can). Others have been outright ignored, despite multiple requests for commentary, such as whether the vectors contain the small genomic AAV hepatic enhancers that Dr Alexandra's group has identified. If so, they should probably mention this in the methods section. The comments here are particularly relevant for the synapsin promoter studies discussed in the last rebuttal. Despite this not being as a complete a story as expected from a group with such deep expertise in gene therapy, it has evolved and my initial impression that this paper represents an important practical demonstration of AAV gene therapy for Asl deficiency, a condition that features hyperammonaemia as well as NO axis CNS perturbations (that have been well described here for the first time), still stands. The manuscript has been significantly improved and aside from very minor corrections, such as the language surrounding conditional correction, is suitable to move forward in the publication process.

We apologise for omitting detail concerning small genomic enhancers and have added text, appropriately. The vector genomes used in this manuscript did contain some sequence of the liver-specific enhancer-promoter identified by Alexander's group. This has been added into the Method section. However, we are confident that this does not compromise the conclusions of the manuscript.

We continue to contest the assertion that a clean expressing CNS AAV vector can be constructed (i.e one which achieves specificity after intravenous injection). We would very much appreciate if the reviewer would share specific details of this vector for us, and other researchers, to validate. It would be a very useful vector for many studies.

Reviewer 2

The authors have addressed the concerns of this reviewer. While the manuscript and data could have been improved from a design perspective, I think the data support the general conclusions.

We thank the reviewer for his time reviewing this manuscript and the valuable modifications requested.